# Functional genomics reveals strain-specific genetic requirements conferring hypoxic growth in *Mycobacterium intracellulare*

Yoshitaka Tateishi[1,2]*, Yuriko Ozeki[1], Akihito Nishiyama[1], Yuta Morishige[3], Yusuke Minato[4], Anthony David Baughn[5], Sohkichi Matsumoto[1,6,7,8]

[1]Department of Bacteriology, Graduate School of Medicine and Dental Sciences, Niigata University, Niigata, Japan; [2]Department of Microbiology, Fukushima Medical University, Fukushima, Japan; [3]Department of Mycobacterium Reference and Research, The Research Institute of Tuberculosis, Japan Anti-Tuberculosis Association, Kiyose, Japan; [4]Center for Infectious Disease Research, Fujita Health University, Toyoake, Japan; [5]Department of Microbiology and Immunology, University of Minnesota Medical School, Minneapolis, United States; [6]Medical Microbiology, Faculty of Medicine, Universitas Airlangga, Surabaya, Indonesia; [7]Department of Bacteriology, Osaka Metropolitan University Graduate School of Medicine, Osaka, Japan; [8]Division of Research Aids, Hokkaido University Institute for Vaccine Research and Development, Sapporo, Japan

*For correspondence:
y-tateishi@med.niigata-u.ac.jp

Competing interest: The authors declare that no competing interests exist.

## eLife Assessment

This study makes a **valuable** contribution by elucidating the genetic determinants of growth and fitness across multiple clinical strains of Mycobacterium intracellulare, an understudied non-tuberculous mycobacterium. Using transposon sequencing (Tn-seq), the authors identify a core set of 131 genes essential for bacterial adaptation to hypoxia, providing a **convincing** foundation for anti-mycobacterial drug discovery.

**Abstract** *Mycobacterium intracellulare* is a major etiological agent of the recently expanding *Mycobacterium avium–intracellulare* complex pulmonary disease (MAC-PD). Therapeutic regimens that include a combination of macrolides and antituberculous drugs have been implemented with limited success. To identify novel targets for drug development that accommodate the genomic diversity of *M. avium–intracellulare*, we subjected eight clinical MAC-PD isolates and the type strain ATCC13950 to genome-wide profiling to comprehensively identify universally essential functions by transposon sequencing (TnSeq). Among these strains, we identified 131 shared essential or growth-defect-associated genes by TnSeq. Unlike the type strain, the clinical strains showed increased requirements for genes involved in gluconeogenesis and the type VII secretion system under standard growth conditions, the same genes required for hypoxic pellicle-type biofilm formation in ATCC13950. Consistent with the central role of hypoxia in the evolution of *M. intracellulare*, the clinical MAC-PD strains showed more rapid adaptation to hypoxic growth than the type strain. Importantly, the increased requirements of hypoxic fitness genes were confirmed in a mouse lung infection model. These findings confirm the concordant genetic requirements under hypoxic conditions in vitro and hypoxia-related conditions in vivo and highlight the importance of using clinical strains and host-relevant growth conditions to identify high-value targets for drug development.

## Introduction

In contrast to the ongoing decline in the rate of tuberculosis, *Mycobacterium avium–intracellulare* complex pulmonary disease (MAC-PD) is increasing in many parts of the world (*Adjemian et al., 2012*; *Namkoong et al., 2016*). The standard therapy for MAC-PD involves multidrug chemotherapy that includes clarithromycin and several antituberculous drugs. Unfortunately, treatment failure is common, resulting in relapse, disease progression, and death (*Daley et al., 2020*). Therefore, it is critical to investigate the bacterial physiological characteristics that underlie the ability of MAC-PD strains to establish and maintain chronic drug-recalcitrant infections.

Recent advances in comparative genomics have revealed major differences in the genomic features of reference type strains and circulating pathogenic clinical isolates. Extensive genomic diversity has been recognized across clinical isolates of nontuberculous mycobacteria (NTM). *Mycobacterium avium* has been divided into four subspecies, including subspecies *avium*, *hominissuis*, *silvaticum*, and *para-tuberculosis* (*Uchiya et al., 2017*). Whereas *M. intracellulare* has been divided into two subgroups, typical *M. intracellulare* (TMI) and *M. paraintracellulare–M. indicus pranii* (MP-MIP), with currently two additional distinguished subspecies, *yongonense* and *chimaera* (*Tateishi et al., 2021*; *Tortoli et al., 2019*). Despite the increasing number of sequenced MAC-PD strains, the molecular genetic mechanisms that underlie their virulence and pathogenesis are still poorly understood.

The burgeoning field of functional genomics has allowed the rapid assessment of molecular aspects of organisms on a genome-wide scale. Transposon (Tn) sequencing (TnSeq) is a functional genomic approach that combines saturation-level transposon-insertion mutagenesis with next-generation sequencing (NGS) to comprehensively evaluate the fitness costs associated with gene inactivation in bacteria. Over the last decade, TnSeq has been applied to numerous bacterial species using various approaches (*van Opijnen and Camilli, 2013*). Whereas initial studies applied TnSeq to reference strains, recent approaches have included the comparative assessment of representative clinical strains to better understand the fundamental aspects of pathogen biology (*Carey et al., 2018*; *Akusobi et al., 2025*). We have previously used TnSeq to globally characterize the genes that are essential for growth versus those that are essential for hypoxic pellicle formation in the *M. intracellulare* type strain ATCC13950 (*Tateishi et al., 2020*). However, given the recent expansion of MAC-PD and the well-recognized high levels of genomic diversity in this complex of bacteria, data obtained from the study of a decades-old reference strain is not expected to represent the diversity of extant clinical isolates. ATCC13950 was originally isolated from the abdominal lymph node of the 34-month-old female (*Cuttino and McCABE, 1949*). Even though the etiological bacterial species are the same, the clinical manifestation of infant lymphadenitis differs markedly from MAC-PD, which often occurs in middle-aged and elderly female patients with no predisposing immunological disorder. So far, there are a total of two reports on TnSeq that compare genetic requirements between clinical mycobacterial strains and the type strains in *M. tuberculosis* (*Carey et al., 2018*) and *M. abscessus* (*Akusobi et al., 2025*). They reported the diversity of genetic requirements among strains, including the type strain, such as *M. tuberculosis* H37Rv and *M. abscessus* ATCC19977. Therefore, it is essential to investigate a collection of strains that captures the diversity of recent clinical MAC-PD strains in order to define key molecular aspects of their pathogenesis and virulence.

In this study, we used TnSeq to analyze a set of recently isolated clinical *M. intracellulare* strains with diverse genotypes. We integrated the gene essentiality data for these strains and the type strain to identify the common essential and growth-defect-associated genes that represent the genomic diversity of this group of pathogens. We also identified the greater requirements of genes involved in gluconeogenesis, the type VII secretion system, and cysteine desulfurase in the clinical MAC-PD strains than in the type strain. Furthermore, the requirement for these genes was confirmed in a mouse lung infection model. The profiles of genetic requirements in clinical MAC-PD strains suggest the mechanism of hypoxic adaptation in NTM in terms of promising drug targets, especially for strains that cause clinical MAC-PD.

## Results

### Common essential and growth-defect-associated genes representing the genomic diversity of *M. intracellulare* strains

To obtain comprehensive information on the genome-wide gene essentiality of *M. intracellulare*, we included nine representative strains of *M. intracellulare* with diverse genotypes for TnSeq in vitro in this study (*Figure 1a and b*, *Figure 1—figure supplement 1*, *Supplementary file 1*). We used TRANSIT (*DeJesus et al., 2015*) to obtain 2–6 million reads of Tn insertions. The average number of Tn insertion reads was >50 at Tn-inserted TA sites, which confirmed the TnSeq data (*Supplementary file 2*). The numbers of essential genes in the clinical MAC-PD strains, calculated with the hidden Markov model (HMM; a transition probability model), tended to be smaller than that in ATCC13950 (*Supplementary file 3*). In total, 131 genes were identified as essential or growth-defect-associated with the HMM analysis across all *M. intracellulare* strains, including the eight MAC-PD clinical strains and ATCC13950 (*Figure 2a*, *Supplementary file 4*). The genes identified as universal essential or growth-defect-associated were involved in fundamental functions, such as glyoxylate metabolism, purine/pyrimidine metabolism, amino acid synthesis, lipid biosynthesis, arabinogalactan/peptidoglycan biosynthesis, porphyrin metabolism, DNA replication, transcription, amino acid tRNA ligases, ribosomal proteins, general secretion proteins, components of iron–sulfur cluster assembly, some ABC transporters, and the type VII secretion system. The genes identified corresponded to the genes that encode conventional and under-development drug targets, such as those encoding gyrase, arabinosyl-transferase, enoyl-(acyl-carrier-protein) reductase, alanine racemase, DNA-directed RNA polymerase, malate synthase (*Krieger et al., 2012*), and $F_0F_1$ ATP synthase (*Figure 2b*). These data suggest that the set of universal essential or growth-defect-associated genes identified with the HMM analysis of *M. intracellulare* strains with diverse genotypes has potential utility as antibiotic drug targets.

### The sharing of strain-dependent and accessory essential and growth-defect-associated genes with genes required for hypoxic pellicle formation in the type strain ATCC13950

By comparative genomic analysis, we have revealed 3153 core genes among 55 *M. intracellulare* strains, including those used in this study (*Tateishi et al., 2021*; *Figure 1—figure supplement 1*). Of these core genes, 439 genes were classified as strain-dependent essential or growth-defect-associated with the HMM analysis (*Supplementary file 5*). And among 5824 accessory genes, 222 genes were classified as essential or growth-defect-associated with the HMM analysis. The proportion of these strain-dependent and accessory essential or growth-defect-associated genes was comparable to the similar analysis of *Streptococcus pneumoniae* (*Rosconi et al., 2022*). Of note, 17 (3.8%) of strain-dependent essential or growth-defect-associated genes and 14 (6.3%) of accessory essential or growth-defect-associated genes were also included in the 175 genes required for hypoxic pellicle formation in ATCC13950 reported in our previous study (*Tateishi et al., 2020*; *Supplementary file 6*). Gene set enrichment analysis (GSEA) (*Subramanian et al., 2005*) showed that 9.1% (16 out of 175) genes were hit as core enrichment (*Supplementary file 6*). Of them, four genes were hit commonly as genes showing increased genetic requirements analyzed by resampling plus HMM analyses, including genes of phosphoenolpyruvate carboxykinase *pckA* (OCU_RS48660), type VII secretion-associated serine protease *mycP5* (OCU_RS38275), type VII secretion protein eccC5 (OCU_RS38345) and glycine cleavage system amino-methyltransferase *gcvT* (OCU_RS35955).

### Partial overlap of the genes showing increased genetic requirements in clinical MAC-PD strains with those required for hypoxic pellicle formation in the type strain ATCC13950

The profiles of genetic requirements in each bacterial strain reflect the adaptation to the environment in which each strain lives. When the strains are placed in a special environment, they can adapt to the situation by altering the profiles of genetic requirements, resulting in the remodeling of metabolic pathways. The sharing of strain-dependent and accessory essential or growth-defect-associated genes with genes required for hypoxic pellicle formation in ATCC13950 prompts us to consider that the profiles of genetic requirements in clinical MAC-PD strains may be associated with the genes required for hypoxic pellicle formation in ATCC13950. To clarify this, we compared the genetic

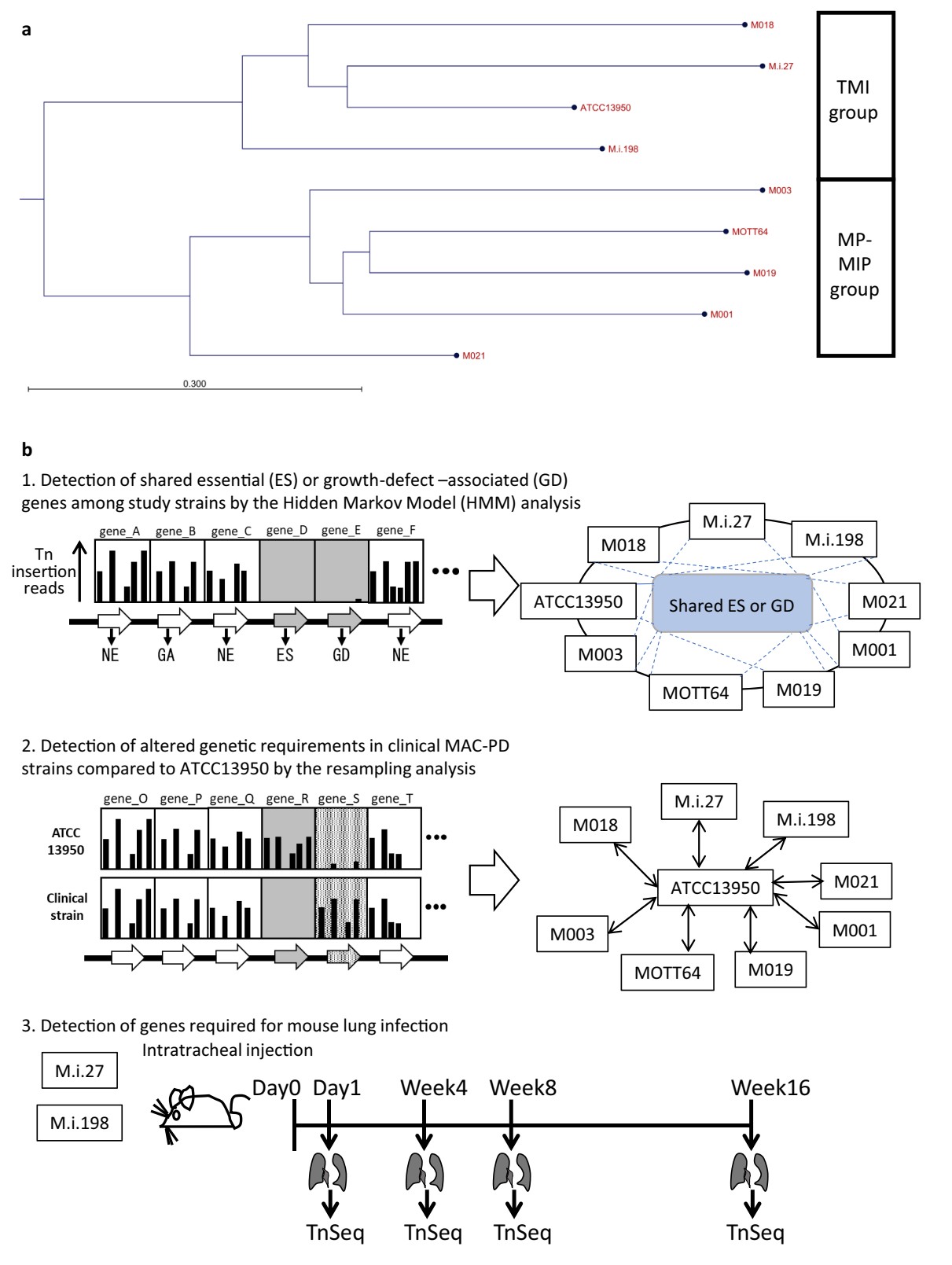

**Figure 1.** Phylogenetic tree of the *M. intracellulare* strains and strategy of the Transposon sequencing (TnSeq) analyses in this study. (**a**) Phylogenetic tree of the *M. intracellulare* strains used in this study. The tree was generated based on average nucleotide identity (ANI) with the neighbor-joining method. TMI: typical *M. intracellulare*; MP-MIP: *M. paraintracellulare–M. indicus pranii*. The detail of the phylogenetic tree is shown in *Figure 1—figure supplement 1*. (**b**) Strategy of the procedure of TnSeq analyses.

*Figure 1 continued on next page*

*Figure 1 continued*

The online version of this article includes the following figure supplement(s) for figure 1:

**Figure supplement 1.** Subject *M. intracellulare* strains in this study.

requirements of the clinical MAC-PD strains with that in ATCC13950 by integrating the HMM analysis with the resampling analysis, a method of calculating the number of significantly altered Tn insertion reads (a gene-based permutation model). With this integrating analysis, 121 genes in the clinical MAC-PD strains were shown to have significantly fewer Tn insertions than ATCC13950 (*Figure 3*, *Supplementary files 7 and 8*). Of these, nine genes were identified in 3–5 clinical MAC-PD strains. These included genes encoding phosphatases (OCU_RS30185, OCU_RS35795), cysteine desulfurase (*csd* [OCU_RS40305]), methyl transferase (OCU_RS41540), a hypothetical protein (OCU_RS47290), type VII secretion components (*mycP5* [OCU_RS38275], *eccC5* [OCU_RS38345]), a glycine cleavage system aminomethyltransferase (*gcvT* [OCU_RS35955]), and phosphoenolpyruvate carboxykinase (*pckA* [OCU_RS48660]) (*Figure 3* and *Figure 4*). The genes involved in gluconeogenesis (such as *pckA*), the glycine cleavage system, and the type VII secretion system were also identified as required for hypoxic pellicle formation in ATCC13950 in our previous study (*Akusobi et al., 2025*). The Tn insertion reads were significantly reduced in the pathways of gluconeogenesis, including in fructose-1,6-bisphosphatase (*glpX*), phosphoenolpyruvate carboxykinase (*pckA*), and pyruvate carboxylase (*pca*), but were significantly increased in those of glycolysis, including in pyruvate kinase (*pykA*) and pyruvate dehydrogenase (*aceE*, *lpdC*), suggesting that carbohydrate metabolism was metabolically remodeled in the clinical MAC-PD strains, which mimics the metabolism observed during hypoxic pellicle formation in ATCC13950 (*Figure 5*).

## Minor role of gene duplication on reduced genetic requirements in clinical MAC-PD strains

On the other hand, 162 genes were detected with significantly more Tn insertions in the clinical MAC-PD strains than in ATCC13950 (*Figure 3*). Of these, 12 genes were identified in more than seven strains. The increased Tn insertion reads in the genes encoding superoxide dismutase (OCU_RS25945) and pyruvate dehydrogenase (*aceE* [OCU_RS35720]) in the clinical MAC-PD strains suggest the inessentiality of the detoxification of oxygen radicals and glycolysis metabolism in these strains (*Figures 3–5*). These genes also included those encoding amidohydrolases (OCU_RS34400, OCU_RS34425, OCU_RS34445), some transcription factors (OCU_RS28440, OCU_RS30230), a CoA transferase (OCU_RS34390), and some predicted transporters (OCU_RS28435, OCU_RS39950, OCU_RS39955, OCU_RS44685). Furthermore, the increase in Tn insertion reads in predicted multidrug-resistant genes, such as those encoding the DrrAB-ABC transporter complex (OCU_RS39950, OCU_RS39955) (*Coll et al., 2018*) and a putative lipoprotein LpqB (OCU_RS44685), encoded by the two-component system *mtrAB* operon (*Nguyen et al., 2010*; *Li et al., 2022*), in the clinical MAC-PD strains were found.

In order to investigate the effect of gene duplication on the change of genetic requirements between strains, we conducted a BLAST search for the 162 genes showing increased Tn insertion reads in the clinical MAC-PD strains compared to ATCC13950. We found that M019 has duplicate genes of OCU_RS44705 coding adenosylhomocysteinase (LOCUS_42940: ahcY_1, LOCUS_21000: ahcY_2). However, there were no duplicate genes found in the remaining 161 genes. These data suggest that gene duplication has minor effects on the change of genetic requirements between strains. Rather, sequence differences and accessory genes may play a key role in determining the difference in genetic requirements.

## Identification of genes in the clinical MAC-PD strains required for mouse lung infection

Hypoxia is the major environmental condition in natural water (*Kirschner et al., 1992*), biofilms (*Werner et al., 2004*), and infections in humans and animals characterized by caseous granuloma lesions (*Via et al., 2008*). The impact of hypoxia on mycobacteria under various ecological circumstances implies that the genes required for pathogenesis of MAC-PD may be, to some degree, overlapped with the genes with increased requirements in the clinical MAC-PD strains compared to ATCC13950, and also

**a**

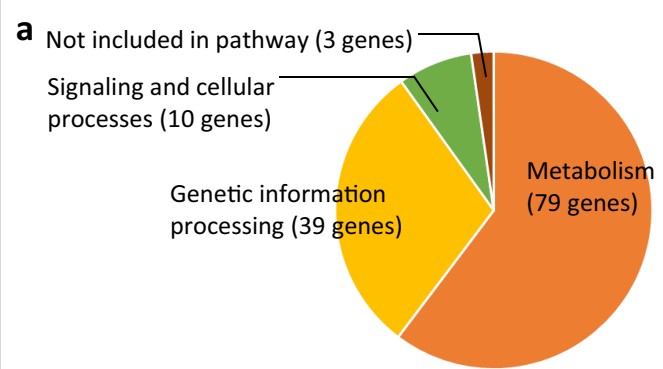

Not included in pathway (3 genes)

Signaling and cellular processes (10 genes)

Genetic information processing (39 genes)

Metabolism (79 genes)

Metabolism: glyoxylate, purine/pyrimidine, amino acid, lipid, arabinogalactan/peptidoglycan, porphyrin

Genetic information processing: DNA replication, transcription, amino acid tRNA ligases, ribosomal proteins, general Sec proteins

Signaling and cellular processes: ABC transporter, type VII secretion system, iron-sulfur cluster assembly

Not Included in Pathway: hypothetical proteins

**b**

| Gene locus | gene | function | target drug |
|---|---|---|---|
| OCU_RS25020, OCU_RS25025 | *gyrB, gyrA* | DNA topoisomerase subunit B, intein-containing DNA gyrase subunit A | fluoroquinolones |
| OCU_RS26150 | *embB* | arabinosyltransferase | ethambutol |
| OCU_RS40185 | *inhA* | enoyl[acyl-carrier-protein] reductase FabI | isoniazid/ethionamid |
| OCU_RS45570 | *alr* | alanine racemase | cycloserine |
| OCU_RS46225 | *rpoC* | DNA-directed RNA polymerase subunit beta' | rifampicin |
| OCU_RS38100 | *glcB* | malate synthase G | malate synthase inhibitors |
| OCU_RS32840, OCU_RS32845, OCU_RS32850 | *atpA, atpG, atpB* | $F_0F_1$ ATP synthase subunit alpha, gamma, beta | bedaquiline |

**c**

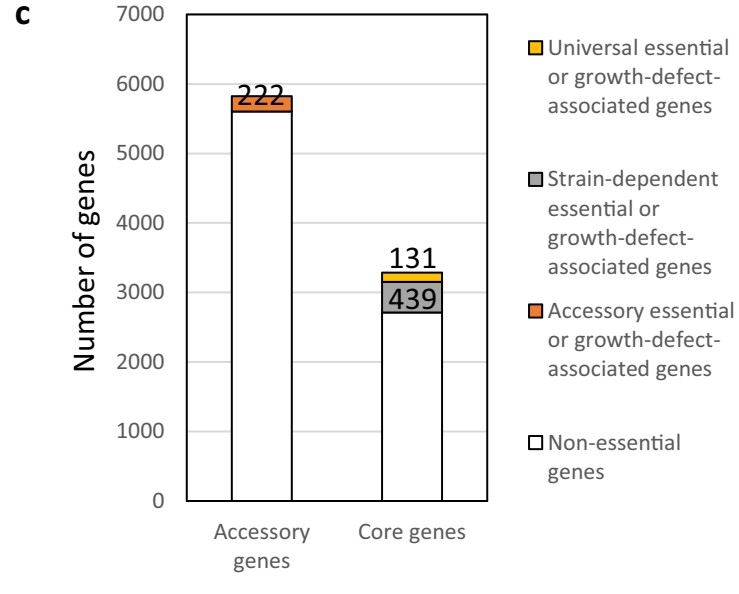

Universal essential or growth-defect-associated genes

Strain-dependent essential or growth-defect-associated genes

Accessory essential or growth-defect-associated genes

Non-essential genes

**Figure 2.** Identification of the essential and growth-defect-associated genes across genetically diverse nine *M. intracellulare* strains used in this study. (**a**) Functional categories of 131 genes identified as universal essential or growth-defect-associated with an hidden Markov model (HMM) analysis. The genes were categorized according to the Kyoto Encyclopedia of Genes and Genomes (KEGG) database. (**b**) Universal essential or growth-defect-associated genes corresponding to the genes of existing antituberculous drug targets. (**c**) The number of the essential or growth-defect-associated genes in the accessory genes, and the number of strain-dependent essential or growth-defect-associated genes in the core genes.

The online version of this article includes the following figure supplement(s) for figure 2:

**Figure supplement 1.** Summary of the gene set enrichment analysis (GSEA) results.

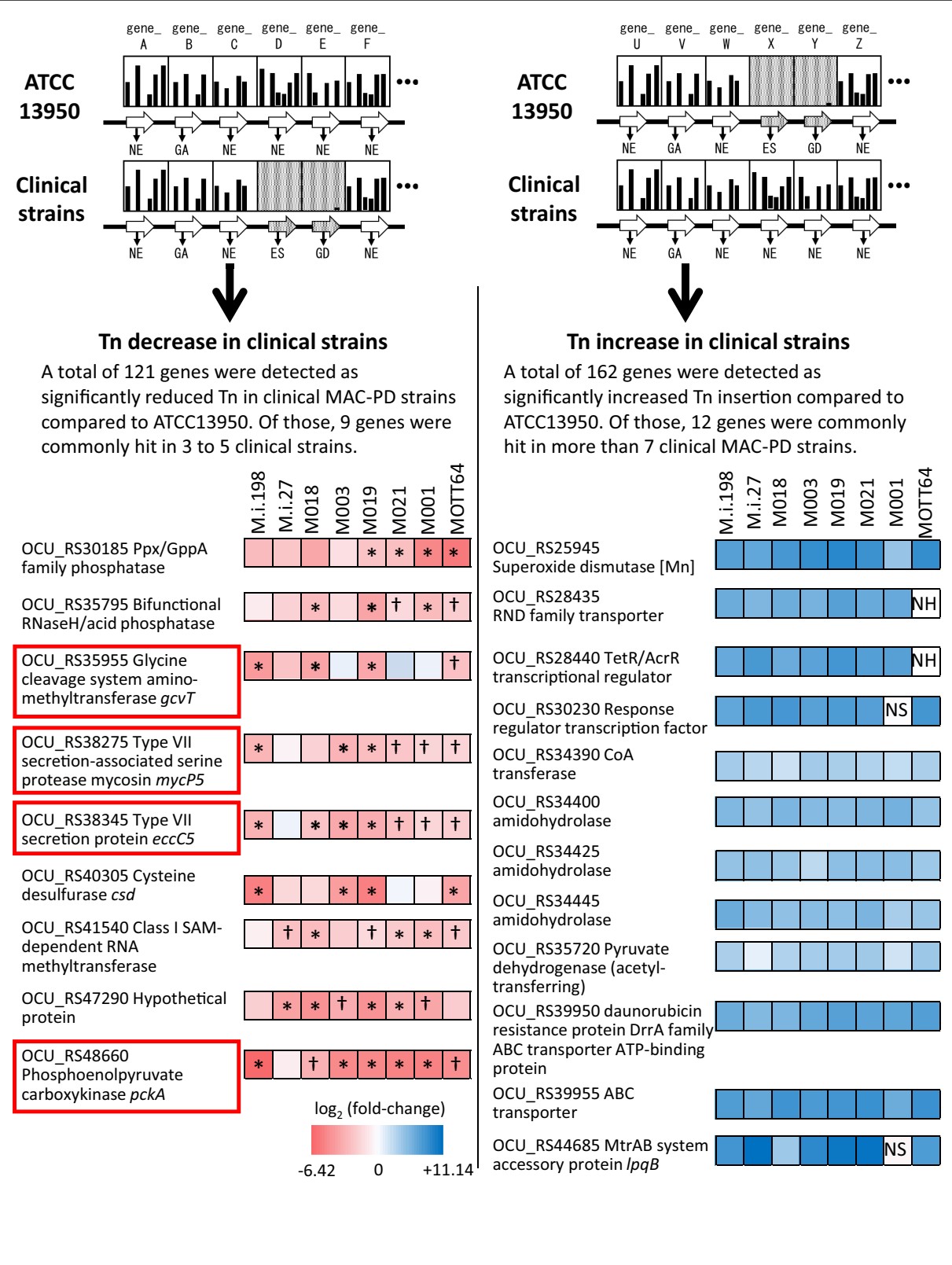

**Figure 3.** Detection of genes showing increased or reduced genetic requirements in the clinical *M. intracellulare* strains. Left panel shows the genes identified as having fewer transposon (Tn) insertion reads than the type strain ATCC13950. The fold changes in the number of Tn insertion reads calculated by a resampling analysis are represented by the color scale. Red squares indicate genes required for hypoxic pellicle formation in ATCC13950 (*Tateishi et al., 2020*). *Genes identified with a combination of resampling and hidden Markov model (HMM) analyses. †Genes identified only with

*Figure 3 continued on next page*

*Figure 3 continued*

resampling analysis. Right panel shows the genes identified as having more Tn insertion reads than the type strain ATCC13950. NH: no homolog with ATCC13950, NS: no significant increase in Tn insertion reads by a resampling analysis (adjusted *p*<0.05).

with the genes required for hypoxic pellicle formation in ATCC13950. To identify genes required for in vivo infection of clinical MAC-PD strains, we performed TnSeq in mice infected intratracheally with the Tn mutant library strains of M.i.198 or M.i.27, which we had demonstrated to be virulent in mice (*Figure 1b*, *Figure 6—figure supplement 1*; *Tateishi et al., 2023*). It is impossible to perform TnSeq in mouse lungs infected with ATCC13950 because ATCC13950 is eliminated within 4 weeks of infection (*Tateishi et al., 2023*). The time course of the changes in the bacterial burden showed a pattern similar to those of the wild-type strains M.i.198 and M.i.27, respectively, except that it was not possible to harvest sufficient colonies (as few as $10^4$ /mouse) in the few mice infected with the M.i.27 Tn mutant strain in Week 8 and Week 16 (*Figure 6—figure supplement 1b*, *Supplementary file 9*). With a resampling analysis, 629 and 512 genes were shown to be required for in vivo infection with transposon mutant library strains M.i.27 and M.i.198, respectively, at least at one time point between Day 1 and Week 16 of infection (*Figure 6a*, *Figure 6—figure supplement 1c and d*, *Supplementary files 10 and 11*). Of these genes, 72 (11.9%) and 50 (9.76%) in M.i.27 and M.i.198, respectively, were

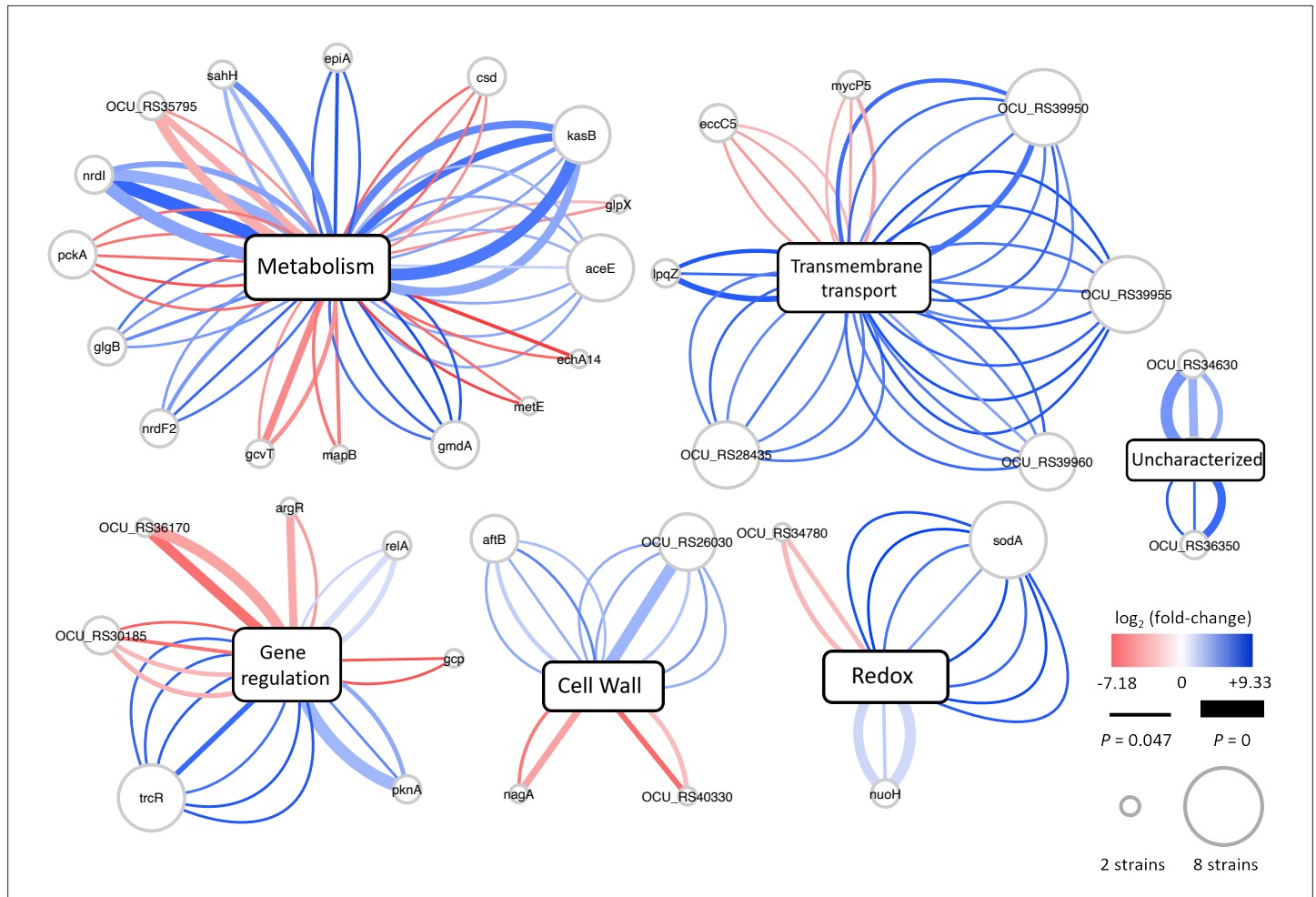

**Figure 4.** Overview of the differences in genetic requirements between the clinical *M. intracellulare* strains and ATCC13950, drawn with Cytoscape (*Shannon et al., 2003*). Each central node represents the functional category of genes, assigned with a Kyoto Encyclopedia of Genes and Genomes (KEGG) pathway analysis. Each peripheral node represents the genes showing significant changes in the number of Tn insertion reads. The size of each node represents the number of strains identified. Each edge represents a strain identified as showing significant changes in the number of transposon (Tn) insertion reads. The thickness of each edge represents the adjusted *p* value. The color scale of each edge represents the fold change in the number of Tn insertion reads calculated by a resampling analysis. The graphical organization was referenced to the previous publication by *Carey et al., 2018*.

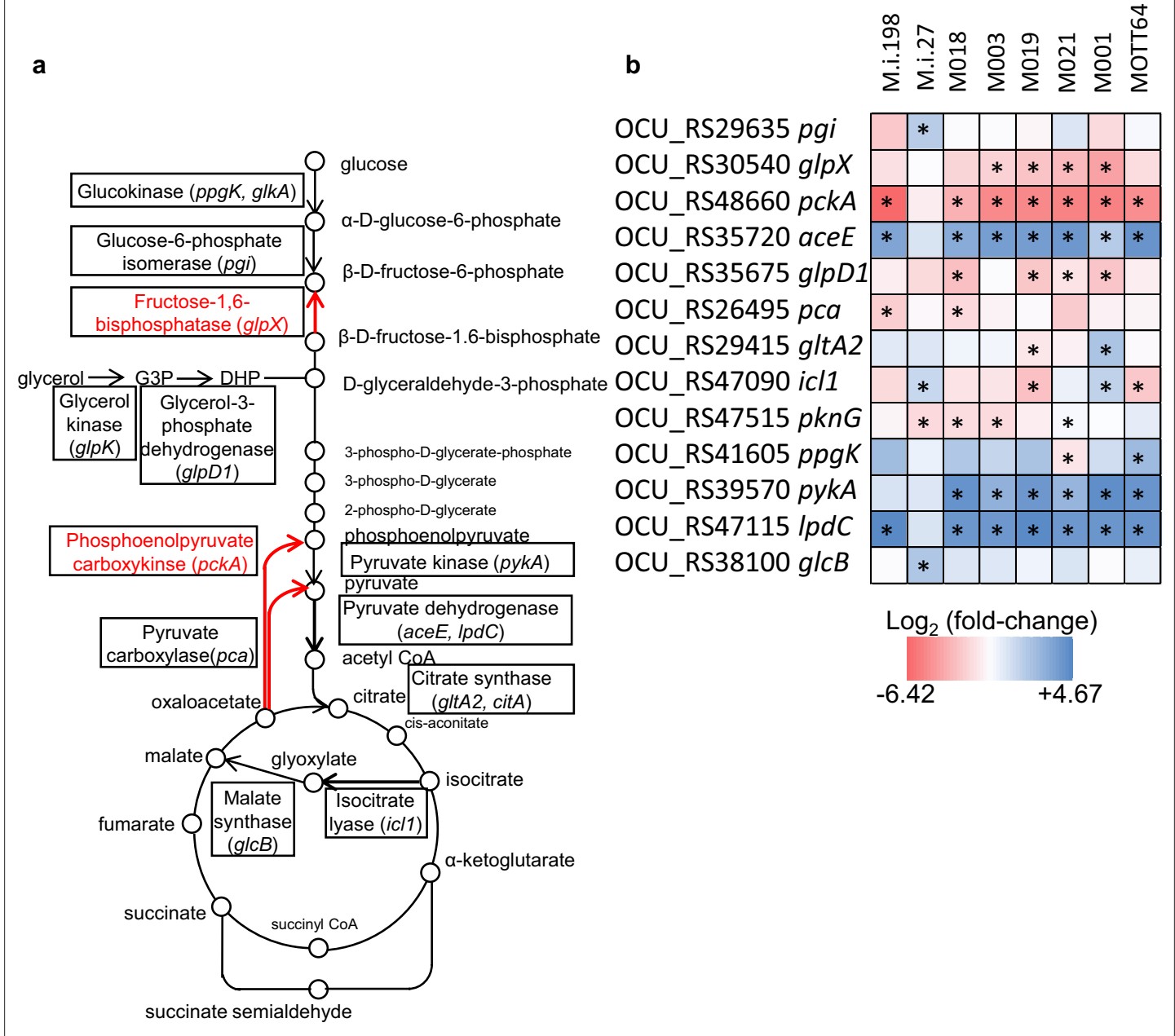

**Figure 5.** Preferential genetic requirements for gluconeogenesis in the clinical *M. intracellulare* strains inferred from the Transposon sequencing (TnSeq) results. (**a**) Carbohydrate pathway showing the changes in Tn insertion reads. Genes identified with the resampling analysis are framed by squares. The essentiality of gluconeogenesis-related genes (*pckA*, *glpX*) was higher and that of glycolysis-related genes (*aceE*, *lpdC*) was lower in the clinical *M. intracellulare* strains compared with those in ATCC13950. (**b**) Data on genetic requirements calculated with a resampling analysis in the clinical *M. intracellulare* strains compared with ATCC13950. Red indicates a reduction in transposon (Tn) insertion reads in the clinical *M. intracellulare* strains. Blue indicates an increase of Tn insertion reads in the clinical *M. intracellulare* strains. Asterisks indicate statistical significance of log₂ fold changes (log₂FCs). The graphical organization was referenced to the previous publication by *Carey et al., 2018*.

also required for hypoxic pellicle formation in ATCC13950. Forty-one genes also required for hypoxic pellicle formation in ATCC13950 were identified in both M.i.27 and M.i.198 and included genes associated with succinate production (multifunctional 2-oxoglutarate metabolism enzyme) and the glyoxylate cycle (isocitrate lyase), cysteine and methionine synthesis (thiosulfate sulfurtransferase [CysA], 5-methyltetrahydrofolate-homocysteine methyltransferase [MetH]), the serine/threonine-protein kinase PknG system, proteasome subunits (PrcA, PrcB), DNA repair UvrC, secretion protein EspR, cell-wall synthesis proteins (such as phosphatidylinositol [polyprenol-phosphate-mannose-dependent

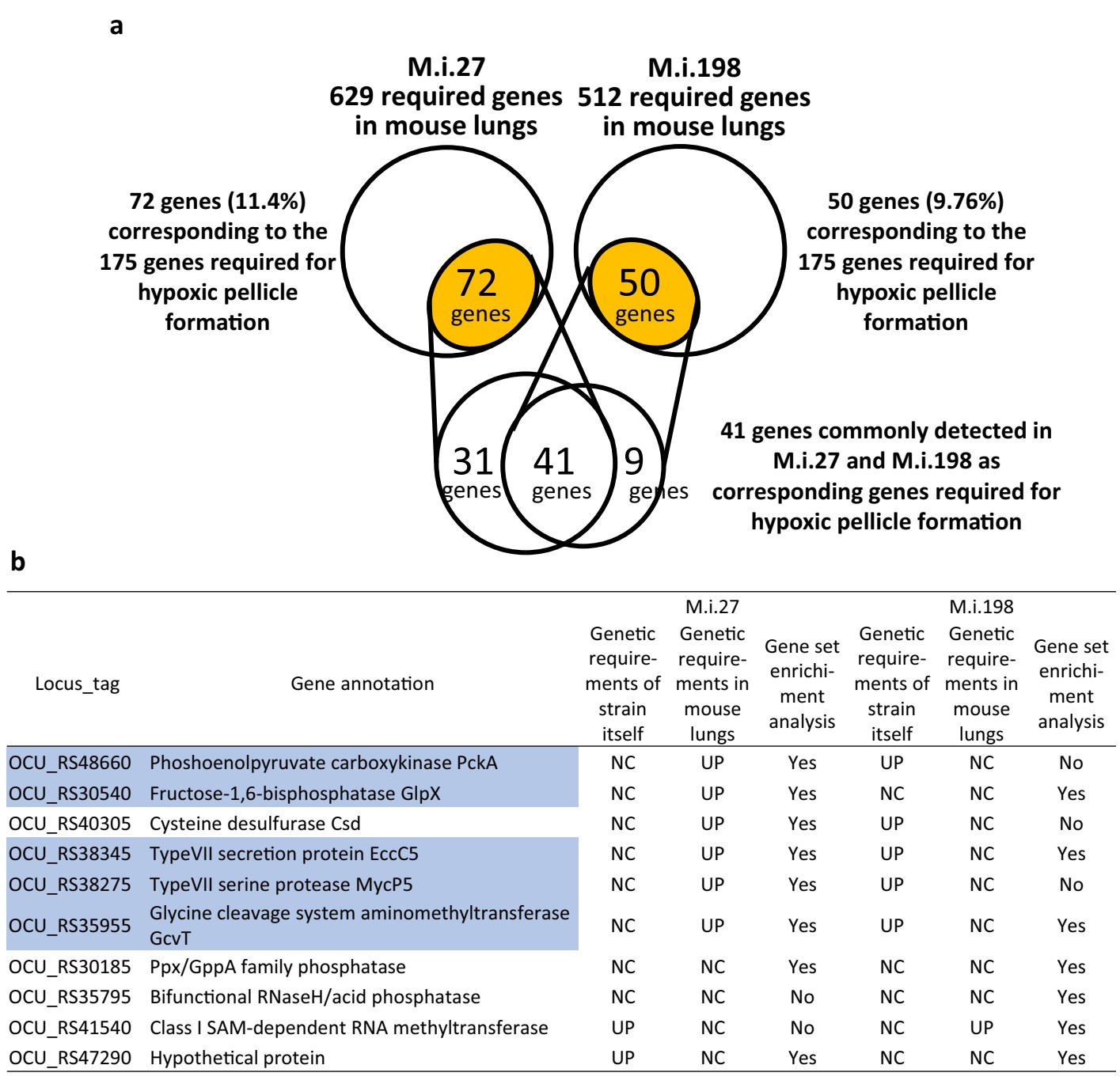

**Figure 6.** Detection of genes required for mouse lung infection in the clinical *M. intracellulare* strains. (**a**) Consistency between the genes required for infection in mouse lungs and the 175 genes required for hypoxic pellicle formation in ATCC13950. (**b**) Changes in the requirements for genes with increased genetic requirements observed in the clinical *M. intracellulare* strains after the infection of mouse lungs. Highlighted genes are also required for hypoxic pellicle formation by ATCC13950. Genetic requirements of the strain itself: genetic requirements compared with that of ATCC13950. Genetic requirements in mouse lungs: changes in the genetic requirements in the infected mouse lungs compared with those before infection. Gene set enrichment analysis: The genes listed as core enrichment are shown as 'Yes' and the genes not listed as core enrichment are shown as 'No'.

The online version of this article includes the following figure supplement(s) for figure 6:

**Figure supplement 1.** Mouse infection experiment for transposon sequencing (TnSeq).

**Figure supplement 2.** Summary of the gene set enrichment analysis (GSEA) results.

**Figure supplement 3.** Summary of the gene set enrichment analysis (GSEA) results.

**Figure supplement 4.** Summary of the gene set enrichment analysis (GSEA) results.

alpha-[1-2]-phosphatidylinositol pentamannoside mannosyltransferase], peptidoglycan (penicillin-binding protein A pbpA), and L,D-transpeptidase 2), putative membrane proteins and transporters (such as the MmpL family, transport accessory protein MmpS3, transport protein MmpL11, vitamin B$_{12}$ transport ATP-binding protein BacA, ABC transporter substrate-binding protein, and ABC transporter permease), and mammalian cell entry (Mce) proteins (OCU_RS48855–48875) (*Supplementary file 12*).

At all timepoints from Day1 to Week 16, there were 74 and 99 genes were shown to be required for lung infection with M.i.27 and M.i.198, respectively. Of them, 21 (28.3%) and 12 (12.1%) genes belonged to the genes involved in hypoxic pellicle formation in the type strain. At timepoints throughout Week 4 to Week 16, except for Day 1, there were 121 and 172 genes were shown to be required for lung infection with M.i.27 and M.i.198, respectively. Of them, 21 (23.1%) and 30 (18.0%) genes belonged to genes involved in hypoxic pellicle formation in the type strain. These hypoxic pellicle-associated genes detected both in M.i.27 and M.i.198 encoded methionine synthesis (MetH), acyl-CoA dehydrogenase, isocitrate lyase, MMPL family transporter (MmpL11) (from Day 1 to Week 16), additionally multifunctional oxoglutarate decarboxylase/dehydrogenase, proteasome subunits (PrcA, PrcB), ABC transporter ATP-binding protein/permease, lipase chaperone (from Week 4 to Week 16) (new *Supplementary files 13 and 14*).

The genes only identified in M.i.27 that were also required for hypoxic pellicle formation in ATCC13950 included genes associated with gluconeogenesis (*pckA*, *glpX*), the glycine cleavage system (*gcvT*), the type VII secretion system (*eccD5*, *espG5*, *eccC5* [OCU_RS38280-OCU_RS38285-OCU_RS38345], *eccC3* [OCU_RS48145]), and D-inositol 3-phosphate glycosyltransferase (*Supplementary file 12*). These three systems, gluconeogenesis, the glycine cleavage system, and the type VII secretion system, also showed increased genetic requirements in 3–5 clinical MAC-PD strains than in ATCC13950 (*Figures 3 and 6*). Although M.i.27 was excluded as a relevant strain showing increased genetic requirements of these three metabolic systems, the increase in genetic requirements for infection in vivo suggests a positive relationship between hypoxic growth in vitro and bacterial growth in vivo. In contrast, M.i.198 already showed increased genetic requirements in these three metabolic systems under aerobic conditions in vitro, so the genes associated with these three systems were not identified as having fewer Tn insertion reads during infection in vivo. Genes only identified in M.i.198 and required for hypoxic pellicle formation in ATCC13950 included genes associated with succinate synthesis (2-oxoglutarate oxidoreductase subunit KorA) or encoding the HTH-type transcriptional repressor KstR and menaquinone reductase (*Supplementary file 12*). As increased genetic requirements in 3–5 clinical strains (excluding M.i.198) compared with ATCC13950, an increased requirement for the RNA methyltransferase OCU_RS41540 was identified in M.i.198 during infection in vivo. These in vivo TnSeq data suggest that, despite the differences in the profiles of the genes required for in vivo infection between strains, an increase in genetic requirements for hypoxic growth in part contributes to the pathogenesis in vivo.

We have performed a statistical enrichment analysis of gene sets by GSEA *Subramanian et al., 2005* to investigate whether the genes required for growth of M.i.27 and M.i.198 in mouse lungs overlap the gene sets required for hypoxic pellicle formation in ATCC13950, as well as the gene sets showing increased genetic requirements observed in the clinical MAC-PD strains listed in *Figure 3*. Of these gene sets, including a total of 180 genes, 54% (98 genes), and 40% (73 genes) were listed as enriched in genes required for mouse infection in M.i.27 and M.i.198, respectively ($p<0.01$ in M.i.27 and $p=0.016$ in M.i.198) (*Figure 6—figure supplement 2*, *Supplementary file 15*). The enriched genes both in M.i.27 and M.i.198 included genes encoding fructose–1,6-bisphosphatase (*glpX*), a type VII secretion protein (*eccC5*), glycine cleavage system aminomethyltransferase (*gcvT*), some phosphatase (OCU_RS30185), and a hypothetical protein (OCU_RS47290) (*Figure 6*). The enriched genes only in M.i.27 included genes of phosphoenolpyruvate carboxylinase (*pckA*), cysteine desulfurase (*csd*), and the ESX-5 type VII secretion component serine protease (*mycP5*). The enriched genes only in M.i.198 included genes encoding some phosphatase (OCU_RS35975) and class I SAM-dependent RNA methyltransferase (OCU_RS41540). Similarly, 40–50% of the genes required for ATCC13950's hypoxic pellicle formation and strain-dependent/accessory essential genes are significantly enriched individually with the genes required for in vivo bacterial growth (*Figure 6—figure supplements 3 and 4*, *Supplementary files 16 and 17*). These data suggest the sharing of the hypoxia-adaptation pathways between in vitro hypoxic pellicle formation and in vivo bacterial growth.

## Effects of knockdown of universal essential or growth-defect-associated genes in clinical MAC-PD strains

To facilitate the functional analysis of the TnSeq-hit genes, it is necessary to construct the genetically engineered strains. Still now, there are very few reports of gene engineering of MAC strains and none of them has been accompanied by the follow-up studies (*Krzywinska et al., 2005*; *Irani et al., 2004*; *Marklund et al., 1995*; *Marklund et al., 1998*; *Mahenthiralingam et al., 1998*; *Hinds et al., 1999*). Moreover, no attempts have been reported to construct the knockdown strains in MAC strains. With an intention to evaluate the effect of suppressing TnSeq-hit genes on bacterial growth, we used the pRH2052/pRH2521 clustered regularly interspaced short palindromic repeats interference (CRISPR-i) plasmids to construct knockdown strains where the expression of the target gene is suppressed (*Singh et al., 2016*; *Choudhary et al., 2015*; *Rock et al., 2017*). Of the nine *M. intracellulare* strains analyzed in this study, seven could be transformed with the vectors for single guide RNA (sgRNA) expression and dCas9 expression. However, the other two strains, M001 and MOTT64, could not be transformed with these CRISPR-i vectors. To confirm the gene essentiality detected with the HMM analysis, we evaluated the consequent growth inhibition in the knockdown strains of representative universal essential or growth-defect-associated genes, including *glcB*, *inhA*, *gyrB*, and *embB*. The knockdown strains showed growth suppression with comparative growth rates of knockdown strains to vector control strains that do not express single sgRNA reaching $10^{-1}$ to $10^{-4}$ (*Figure 7a*).

## Differential effects of knockdown of accessory/strain-dependent essential or growth-defect-associated genes among clinical MAC-PD strains

As representative accessory and strain-dependent essential or growth-defect-associated genes, we evaluated the nine genes showing reduced Tn insertion reads in the 3–5 clinical MAC-PD strains as listed in *Figure 3*, as well as *glpX*, a gene of gluconeogenesis where four MAC-PD strains showed reduced Tn insertion reads in a resampling analysis (*Figure 5b*). The growth was suppressed in the *pckA*-knockdown strains in M.i.198 and M003, in the *glpX*-knockdown strain in M003, and in the cysteine desulfurase gene (*csd*)-knockdown strains in M.i.198 and M003, with comparative growth rates of knockdown strains to vector control strains reaching $10^{-2}$ to $10^{-4}$ (*Figure 7b*). The growth tended to be suppressed in the type VII secretion protein gene (*eccC5*)-knockdown strain in M003 and in the type VII secretion-associated serine protease mycosin gene (*mycP5*)-knockdown strain in M003, with comparative growth rates of knockdown strains to vector control strains reaching less than $10^{-1}$. By contrast, the growth was not suppressed in these knockdown strains in ATCC13950, with comparative growth rates of knockdown strains to vector control strains not reaching $10^{-1}$, which was consistent with the inessentiality of these genes under aerobic conditions in ATCC13950. As for M.i.198 and M003, the consistency between TnSeq data and CRISPR-i data was supported by a correlation between the growth suppression of the knockdown strains and the fitness costs evaluated by a resampling analysis (*Supplementary file 18*).

In contrast to the cases of M.i.198 and M003, the growth of knockdown strains of the accessory and strain-dependent essential genes was not effectively suppressed in the other MAC-PD strains despite being positively detected in TnSeq (*Figure 7b*, *Figure 7—figure supplement 1*). Although the gene expression levels were not suppressed to <1% as reported in *M. tuberculosis* (Mtb) (*McNeil and Cook, 2019*), the suppression of gene expression was confirmed to approximately 20–70% in the knockdown strains of these accessory and strain-dependent essential or growth-defect-associated genes, as well as universal essential or growth-defect-associated genes, such as *glcB*, *inhA*, *gyrB*, and *embB* (*Figure 7—figure supplement 2*). The possible reason may be explained by the recently revealed bypass mechanism of gene essentiality in strain-dependent/accessory essential or growth-defect-associated genes, where gene essentiality can be bypassed by several mechanisms, including the composition of the accessory genome and pathway rewiring (*Rosconi et al., 2022*).

To be summarized, we could validate the universal essential or growth-defect-associated genes by the CRISPR-i system in the *M. intracellulare* strains used in this study. And, although not successful with all strains, we could validate the accessory and strain-dependent essential or growth-defect-associated genes, especially genes of gluconeogenesis, type VII secretion components, and iron-sulfur cluster assembly in several MAC-PD strains like M.i.198 and M003.

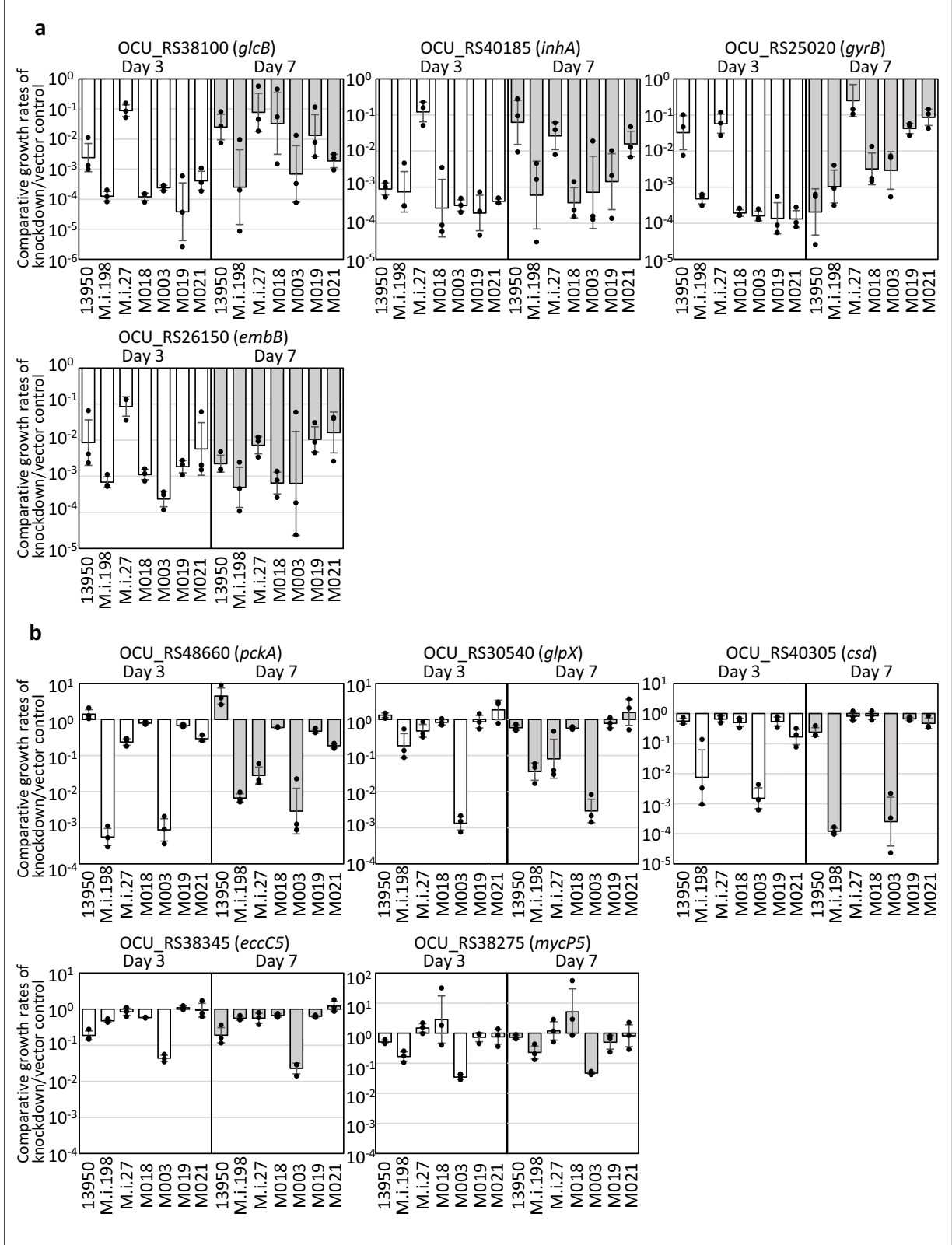

**Figure 7.** Evaluation of the effect of the suppression of Transposon sequencing (TnSeq)-hit genes on the bacterial growth using the CRISPR-i system. Open bar: day 3; closed bar: day 7. (**a**) Comparative growth rates of the knockdown strains relative to those of the vector control strains in the representative universal essential or growth-defect-associated genes: *glcB, inhA, gyrB,* and *embB*. (**b**) Comparative growth rates of the knockdown strains relative to those of the vector control strains in the representative accessory and strain-dependent essential or growth-defect-associated genes:

*Figure 7 continued on next page*

*Figure 7 continued*

*pckA*, *glpX*, *csd,* and ESX-5 type VII secretion components. Data are shown as the means ± SD of triplicate experiments. Data from one experiment representative of two independent experiments (N=2) are shown.

The online version of this article includes the following figure supplement(s) for figure 7:

**Figure supplement 1.** Comparative growth rates of the knockdown strains relative to those of the vector control strains in several accessory and strain-dependent essential or growth-defect-associated genes.

**Figure supplement 2.** Quantitative reverse transcription PCR (qRT-PCR) validation of the suppression of gene expression in knockdown strains of universal and strain-dependent/ accessory essential or growth-defect-associated genes.

## Preferential hypoxic adaptation of clinical MAC-PD strains evaluated with bacterial growth kinetics

The metabolic remodeling, such as the increased genetic requirements of gluconeogenesis and the type VII secretion system, and the overlap of the genes required for mouse lung infection and those required for hypoxic pellicle formation involved by these metabolic pathways suggest the preferential adaptation of the clinical MAC-PD strains to hypoxic conditions. To confirm this, we investigated the growth kinetics of each strain under aerobic and hypoxic conditions. In terms of growth kinetics, most of the clinical MAC-PD strains (except M001) entered logarithmic (log) phase earlier than ATCC13950 under 5% oxygen (*Figure 8a and b*, *Supplementary file 19*). However, under aerobic conditions, the clinical MAC-PD strains entered log phase at a similar time to ATCC13950. The growth rate at the mid-log point (midpoint) was significantly reduced under hypoxic conditions in a total of five clinical MAC-PD strains (M.i.198, M.i.27, M003, M019, and M021) compared to aerobic conditions (*Figure 8a and b*, *Supplementary file 19*). The growth rate at midpoint under hypoxic conditions was significantly lower in these five clinical MAC-PD strains than in ATCC13950. The early entry to the log phase, followed by long-term logarithmic growth (slow growth rate at midpoint), suggests the capacity for these five clinical MAC-PD strains to continue replication for a long time under hypoxic conditions. On the other hand, the remaining three clinical MAC-PD strains (M018, M001, and MOTT64) did not show significant change in the growth rate between aerobic and hypoxic conditions, suggesting that there are different levels of capacity in maintaining long-term replication under hypoxia among clinical MAC-PD strains. In ATCC13950, the entry to log phase was significantly delayed under 5% oxygen compared to aerobic conditions, and the growth rate at midpoint was significantly increased under hypoxic conditions compared to aerobic conditions in ATCC13950. Such a long-term lag phase followed by short-term log phase suggests lower capacity for ATCC13950 to continue replication under hypoxic conditions compared to clinical MAC-PD strains.

## The pattern of hypoxic adaptation is not simply determined by genotypes

Since ATCC13950 and M018 are phylogenetically neighbors within the same TMI group each other, there may be a claim that the growth phenotypes of a longer lag phase compared to other strains under hypoxic conditions might be determined simply by the genotypes. To examine this, we performed hypoxic growth assays for an additional two strains (M005, M016) phylogenetically more closely related to ATCC13950 (*Figure 1—figure supplement 1*, *Supplementary file 1*). The result showed different growth patterns under hypoxia; M005 reached midpoint later than ATCC13950, by contrast M016 reached midpoint three-quarters earlier than ATCC13950 (*Figure 8—figure supplement 1*). These data suggest that genotypes seem to have some impact on bacterial growth pattern under hypoxia; however, since there was a significant difference in the timing of hypoxic adaptation among ATCC13950 and its neighbor strains, bacterial growth pattern under hypoxia seems to be determined by multiple factors other than genotypes, such as strain-specific profiles of genetic requirements and unproven regulatory systems.

In summary, the data on more rapid adaptation to hypoxia followed by reduced growth rate once adapted indicate the greater metabolic advantage on continuous logarithmic bacterial growth under hypoxic conditions in the clinical MAC-PD strains than in ATCC13950. Because the profiles of genetic requirements reflect the adaptation to the environment in which bacteria habits, it is reasonable to assume that the increase of genetic requirements in hypoxia-related genes, such as gluconeogenesis

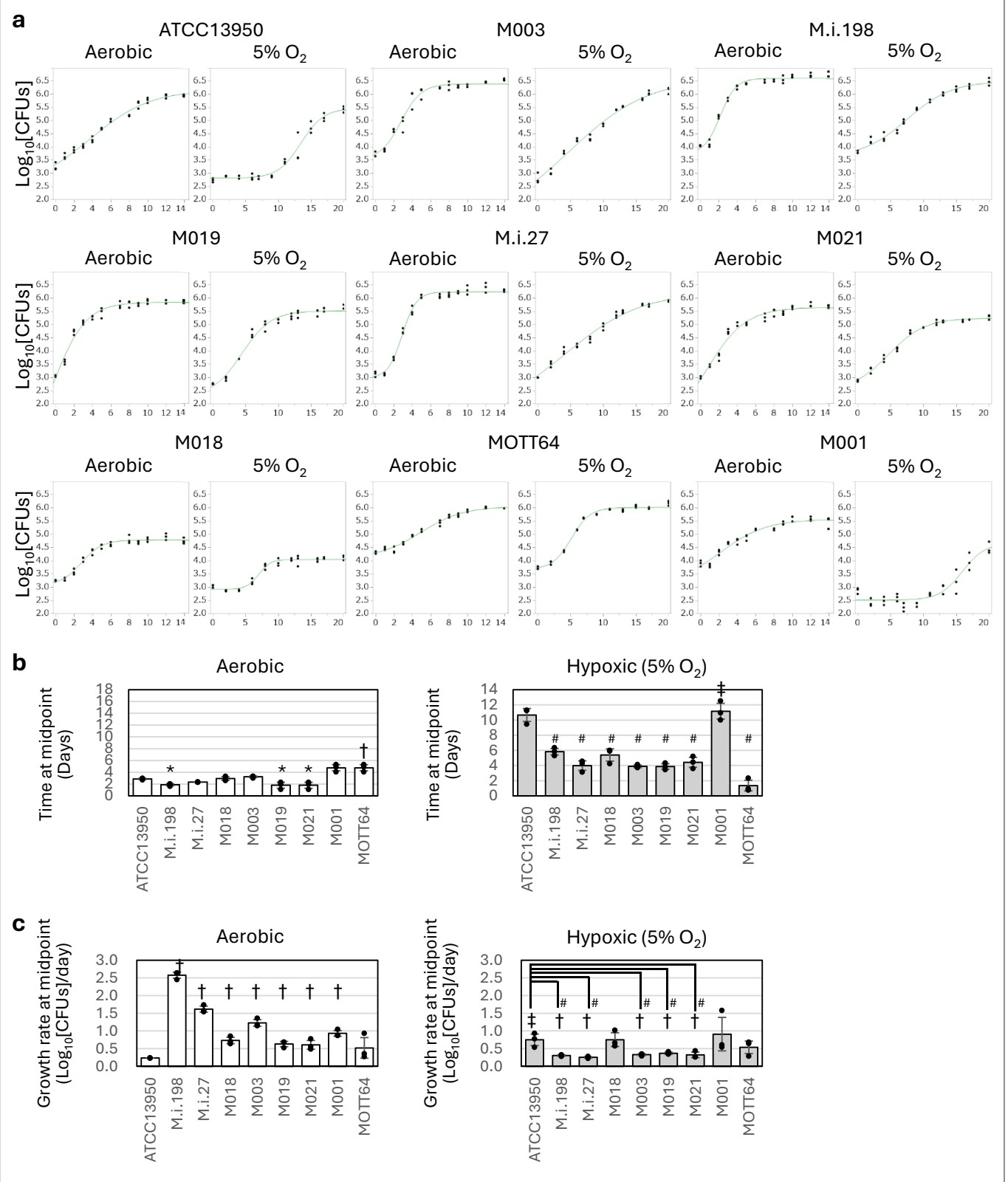

**Figure 8.** Comparison of the timing of entry into logarithmic growth and logarithmic growth rate by the clinical *M. intracellulare* strains and ATCC13950. (**a**) Representative data on the growth curves of each strain under aerobic and 5% oxygen conditions. The assay was performed three times in 96-well plates containing 250 µl of broth medium per well, in triplicate. Data are represented as colony-forming units (CFUs) in 4 µl samples at each timepoint. Data on the growth curves are the means of three biological replicates from one experiment. Data from one experiment representative of three

*Figure 8 continued on next page*

*Figure 8 continued*

independent experiments (N=3) are shown. (**b**) Time at the inflection point (midpoint) on the sigmoid growth curve. * Significantly earlier than an aerobic culture of ATCC13950; # Significantly earlier than a hypoxic culture of ATCC13950; † Significantly later than an aerobic culture of ATCC13950; ‡ Significantly later than a hypoxic culture of ATCC13950. (**c**) Growth rate at midpoint of the growth curve in each strain. # Significantly slower than a hypoxic culture of ATCC13950; †Significantly slower than an aerobic culture of the corresponding strain; ‡Significantly faster than an aerobic culture of ATCC13950. Open bars: aerobic; closed bars: 5% $O_2$. Data are shown as the means ± SD of triplicate experiments. Data from one experiment representative of three independent experiments (N=3) are shown.

The online version of this article includes the following figure supplement(s) for figure 8:

**Figure supplement 1.** Data on the growth curve in ATCC13950 and its neighbor clinical *M. intracellulare* strains M005 and M016.

(*pckA*, *glpX*), type VII secretion system (*mycP5*, *eccC5*), and cysteine desulfurase (*csd*) plays an important role in the growth under hypoxia-relevant conditions in vivo.

## Discussion

In this study, using TnSeq of nine genetically diverse *M. intracellulare* strains, we have identified 131 shared essential and growth-defect-associated genes with an HMM analysis. The genes identified are promising drug targets, representing the genomic diversity of the clinical MAC-PD strains and reflecting the actual clinical situation. The strategy of narrowing the essential genes of bacteria is a useful approach for refining the database of drug targets. We have also demonstrated an overview of the differences in genetic requirements for hypoxic growth between the ATCC type strain and the clinical *M. intracellulare* strains responsible for MAC-PD, with the evidence of earlier entry to log phase followed by slower growth rate observed in most of these clinical strains compared to ATCC13950. This is the first study to demonstrate the positive relationship between the profiles of increased genetic requirements for hypoxic metabolic remodeling and the phenotypes advantageous for bacterial survival under hypoxic conditions.

In this study, 121 genes were detected as significantly reduced Tn insertion in the clinical MAC-PD strains compared to ATCC13950, which was higher than the case of Mtb showing 10–50 genes with altered Tn insertion reads detected by a resampling analysis (*Carey et al., 2018*). This may reflect the genomic diversity of *M. intracellulare* compared with that of Mtb. The proportion of accessory genes is larger in NTM than in Mtb (*Yang et al., 2018*). In a clinical trial of phage therapy for *M. abscessus* infection, the number of accessory genes was shown to decrease with the loss of virulence in *M. abscessus* during the course of therapy (*Nick et al., 2022*), suggesting some role for accessory genes in its clinical pathogenesis. Our data on the overlap of the accessory essential or growth-defect-associated genes, such as *pckA* and *mycP5* (*Supplementary file 7*) with genes required for mouse lung infection (*Figure 6b*, *Figure 6—figure supplement 1*) supports the idea that accessory genes (non-core genes) may encode functions required for bacterial survival, especially under non-optimal conditions, such as hypoxia and during the infection of human and animal bodies.

Our TnSeq data are consistent with metabolomic data that have shown the importance of the gluconeogenic carbon flow of TCA cycle intermediates for bacterial growth under hypoxic conditions in terms of the essentiality of PckA and the reverse TCA cycle (*Eoh et al., 2017*; *Lim et al., 2021*; *Marrero et al., 2010*; *Watanabe et al., 2011*). Gluconeogenesis is essential for bacterial growth when fatty acids are used as a major nutrient (*Marrero et al., 2010*). PckA catalyzes the conversion of the metabolites derived from the TCA cycle to the metabolites of gluconeogenesis (*Lim et al., 2021*). The reaction catalyzed by PckA is essential for bacterial growth when the available nutrients are predominantly fatty acids (*Gouzy et al., 2021*; *Quinonez et al., 2022*). The main nutrients for mycobacteria inside hypoxic granulomas in an infected human or mouse body are lipids, such as fatty acids and cholesterol, rather than carbohydrates (*Marrero et al., 2010*; *Pandey and Sassetti, 2008*). In the present study, *M. intracellulare* grew under 5% oxygen, whereas Mtb stops growing under 1% oxygen and enters a dormant state (*Lim et al., 2021*). The depletion of phosphoenolpyruvate in the *pckA* deletion mutant of Mtb reduced the phosphoenolpyruvate–carbon flux essential for bacterial growth and drug sensitivity (*Lim et al., 2021*). The present study supports the idea that PckA is essential for the microaerobic growth of mycobacteria. Our TnSeq data imply the preferential adaptation of the clinical MAC-PD strains for circumstances in vivo, which are characterized by hypoxia.

On the other hand, such metabolic shift towards gluconeogenesis and reverse TCA cycle can also be observed under reactive nitrogen species and in vivo infection as well. This suggests that the metabolic shift observed under hypoxia is a general stress response rather than being specific to hypoxia due to the nature of the provided/available carbon source (*Prosser et al., 2017*). Several factors other than hypoxia can be estimated to be related to the change of the genetic requirements in the clinical MAC-PD strains, such as host-related factors and general stress conditions (i.e. nutrition of abundant fatty acids and less carbohydrates, low pH circumstances) (*Marrero et al., 2010*; *Gouzy et al., 2021*; *Quinonez et al., 2022*; *Pandey and Sassetti, 2008*). Except for our data of the profile of genes required for hypoxic pellicle formation in *M. intracellulare* (*Tateishi et al., 2020*), no fundamental genome-wide data are available on the change of the genetic requirements in response to each stress factor in non-tuberculous mycobacteria. Further studies are needed to discriminate which factors are specifically involved in the increased requirements of each gene. Although of interest, it is beyond the scope of the current study.

As in the *glpX*-knockdown strains, *csd* knockdown caused growth suppression in several MAC-PD strains. This type II Csd is involved in the biosynthesis of the iron–sulfur clusters of the SUF machinery. Compared with the type I cysteine desulfurase IscS, the SUF machinery is active under oxidative-stress and iron-limiting conditions (*Takahashi and Tokumoto, 2002*). Representatives of iron–sulfur cluster proteins include components of the electron-transport system, such as NADH dehydrogenase I (Nuo) and succinate dehydrogenase (Sdh), which are components of respiratory complexes I and II, respectively. Under hypoxic conditions, the non-proton-translocating type II NADH dehydrogenase Ndh is upregulated, and cytochrome *bd* oxidase (CydAB) and the proton-pumping type I NADH dehydrogenase (Nuo) are downregulated (*Prosser et al., 2017*). In the reverse TCA cycle on the same reaction pathway as Sdh, fumarate reductase (Fnr) is also an iron–sulfur cluster protein that produces succinate and is a proposed alternative electron acceptor under hypoxic conditions (*Watanabe et al., 2011*; *Cecchini et al., 2002*). The increased essentiality of the *csd* gene observed in the clinical MAC-PD strains seems to reflect the switching of the electron-transport system to less-energy-efficient respiration to support preferential hypoxic survival, including in vivo.

About 10% of the genes required for mouse infection are also required for hypoxic pellicle formation. The genes showing increased genetic requirements in the clinical MAC-PD strains, such as *pckA* and genes encoding the ESX-5 type VII secretion system, including *eccC5* and *mycP5*, also showed increased genetic requirements in mouse infections. In a TnSeq study of mice infected intravenously with *M. avium*, the genes of the ESX-5 type VII secretion system were required for infection of the liver and spleen (*Dragset et al., 2019*). The ESX-5 type VII secretion system is involved in the transport of fatty acids and the virulence of mycobacteria (*Ates et al., 2015*; *Ates et al., 2016*). These similarities in the profiles of genetic requirements prompt us to propose some role of biofilm formation in mycobacterial pathogenesis in animals and humans. In the present study, we shed light on the impact of hypoxia on biofilm formation, the acquisition of virulence, and the in vivo pathogenesis of NTM in terms of global genetic requirements. Although the morphologies of the pellicle and macroscopic lesions (including cavitation and nodular bronchiectasis) are not similar, monitoring the expression of the genes identified in this study may provide clues to the mechanism of biofilm formation in vivo.

The data on the validation of the universal essential or growth-defect associated genes by the CRISPR-i system was overall acceptable in the clinically MAC-PD strains whose knockdown strains could be constructed (*Figure 7a*). However, we have encountered the difficulties of validating the TnSeq-hit genes by CRISPR-i experiment, especially accessory and strain-dependent essential or growth-defect-associated genes in the MAC-PD strains other than M.i.198 or M003 (*Figure 7b*, *Figure 7—figure supplement 1*). We found the lower efficiency of the suppression of gene expression by CRISPR-i in *M. intracellulare* than in Mtb that is reported to be suppressed to 6% under 10 ng/ml anhydrotetracycline (aTc) and by <1% under 100 ng/ml aTc in the knockdown strains, compared to strains expressing scrambled single guide RNAs (sgRNA) (*McNeil and Cook, 2019*). However, because the levels of knockdown efficiency were comparable between strain-dependent/accessory essential genes and universally essential genes (*Figure 7—figure supplement 2*), the low efficiency of knockdown does not seem to explain the reason for the discrepancy between TnSeq and CRISPR-i results. Although it is difficult to find the clear definitive reason for this phenomenon only from the current study, the possible reason may be speculated by the bypass mechanism of gene essentiality which is characteristically observed in strain-dependent/accessory essential or growth-defect-associated genes.

According to the publication by *Rosconi et al., 2022* reporting the 'forced-evolution experiments' of 36 clinical *Streptococcus pneumoniae* strains, gene essentiality can be bypassed by several mechanisms, including the composition of the accessory genome and pathway rewiring. They successfully recovered knockout mutants from transformation experiments in strain-dependent/accessory essential genes, such as cytidine monophosphate kinase, a folate pathway enzyme formate tetrahydrofolate ligase and an undecaprenyl phosphate-biosynthesis pathway enzyme farnesyl-diphosphate synthase. The bypassing of gene essentiality could be suggested by observing suppressor mutations and synthetic lethality in knockout strains. By contrast, universal essential genes were reported to fulfill the three categories, including high levels of conservation within and often across species, limited genetic diversity, and high and stable expression levels. Consequently, universal essential genes were estimated to be rigid, largely immutable key components to an organism's survival. The knockout recovery of the universal essential genes was reported to fail, as shown by no colonies or only the appearance of merodiploids. Taking into consideration such bypass mechanism of gene essentiality, the inconsistent effect of gene silencing of strain-dependent/accessory essential genes on bacterial growth may reflect some kinds of pathway rewiring that helps the bacteria grow under suppression of the target gene expression. Nevertheless, our study has opened the avenue for functional analysis of NTM that leads to the development of novel drugs targeting universal and accessory/strain-dependent essential metabolic pathways, especially by making use of M.i.198 and M003 as optimal NTM strains for gene manipulation.

In this study, we revealed the pattern of hypoxic adaptation characteristic to the clinical MAC-PD strains by the growth curve analysis: an early entry to log phase followed by a reduced growth rate (*Figure 8*). This pattern implicates the continuous impact on the infected hosts during logarithmic bacterial growth, which may be involved in the persistent and steadily progressive illness for years in humans (*Daley et al., 2020*). The advantage of slow growth in mycobacteria may be inferred by the role of a histone-like nucleoid protein, mycobacterial DNA-binding protein 1 (MDP1), also designated HupB (*Savitskaya et al., 2018*; *Enany et al., 2017*; *Shaban et al., 2023*). The expression of MDP1 is enhanced in the stationary phase, resulting in reduced growth speed and long-term survival with increased resistance to reactive oxygen species (ROS) (*Savitskaya et al., 2018*; *Enany et al., 2017*; *Shaban et al., 2023*). The reduced growth rate after entry to the log phase may allow long-term survival of mycobacteria under ROS-rich circumstances inside host cells. The pattern of hypoxic adaptation as observed in the clinical MAC-PD strains may enhance pathogenesis by extending the period of the repeated process of bacterial invasion and replication in infected macrophages.

Contrary to our expectation, not all clinical strains showed rapid adaptation from aerobic to hypoxic conditions relative to ATCC13950. The two strains belonging to the MP-MIP subgroup, MOTT64 and M001, showed similar time at midpoint under aerobic conditions. However, the time at midpoint was significantly different between MOTT64 and M001 under hypoxia, the latter showing great delay of timing of entry to logarithmic phase. In contrast to the majority of the clinical strains that showed slow growth at midpoint under hypoxia, neither strain showed such a phenomenon. These data suggest that some, but not a major proportion of, clinical *M. intracellulare* strains do not have an advantageous phenotype for continuous logarithmic growth under hypoxia. In Mtb, the adaptation to hypoxia is considered to be necessary for clinical pathogenesis because the conditions in vivo, such as inside macrophages or granulomas, are hypoxic (*Prosser et al., 2017*; *Rustad et al., 2009*). Our inability to construct knockdown strains in M001 and MOTT64 prevented us from clarifying the factors that discriminate against the pattern of hypoxic adaptation. Although the implication in clinical situations has not been proven, strains without slow growth under hypoxia may have different (possibly strain-specific) mechanisms of hypoxic adaptation corresponding to their growth phenotypes under hypoxia. Nevertheless, because we have shown that most of the *M. intracellulare* strains with different genotypes showed preferential adaptation to hypoxia in this study, our results largely reflect the clinical situation of the etiological agents of MAC-PD. A possible strategy for identifying factors that confer clinical pathogenesis will be to focus on the major clinical strains that show preferential hypoxic adaptation.

We tried to use inhibitors of phosphoenolpyruvate carboxykinase to confirm the increased essentiality of the *pckA* gene. However, we detected no positive effect because millimolar 3-mercaptopicolinic acid is required to show the effect of inhibition, even in eukaryotic cells, which have no cell wall (*Brearley et al., 2020*). Moreover, 3-alkyl-1,8-dibenzylxanthine (*Montal et al., 2019*) also only

effectively inhibits PckA in eukaryotic cells. However, PckA has been proposed as a drug target for Mtb (*Mcleod et al., 2019*), so our TnSeq data seem to be consistent with studies directed towards antimycobacterial drug discovery.

We found the difference in capacity of transformation between strains belonging to the same genomic *M. paraintracellulare-M. indicus pranii* (MP-MIP) subgroup. Although the direct mechanism determining the competency for foreign DNA has not been elucidated in *M. intracellulare* and other NTM species, several studies on general bacteria suggest the difficulties of introducing foreign DNA into clinical strains compared to the laboratory strains. As suggested in *Staphylococcus aureus* (*Corvaglia et al., 2010*), some clinical strains develop elimination systems of foreign nucleic acids, such as a type III-like restriction endonuclease. As suggested in gram-negative bacteria (*Qin et al., 2022*), there may be some difference in cell surface structures between strains, resulting in the necessity of polymyxin B nonapeptide targeting cell membrane for transforming clinical strains. The efficiency of eliminating foreign DNA may be attributed to various kinds of strain-specific factors, including restriction endonuclease, natural CRISPR-interference system, and cell wall structures rather than a simple genotype factor.

There may be a claim that the saturation of Tn insertion in most of each replicate were 50–60% in our libraries. However, our method of constructing Tn mutant libraries is in accordance with the previous studies of 'high-density' transposon libraries (*Akusobi et al., 2025*) because the saturation of the Tn mutant libraries by combining replicates are 62–79% as follows: ATCC13950: 67.6%, M001: 72.9%, M003: 63.0%, M018: 62.4%, M019: 74.5%, M.i.27: 76.6%, M.i.198: 68.0%, MOTT64: 77.6%, M021: 79.9%. That is, we predicted gene essentiality from the Tn mutant libraries with 62–79% saturation in each strain. However, there is a non-permissive sequence pattern in mariner transposon mutagenesis and using more than 10 replicates of Tn mutant libraries is reported to be a more accurate method (*DeJesus et al., 2017*; *Rifat et al., 2021*). The issue of high probability of judging nonessential regions as mistakenly as 'essential' are mainly found in small ORFs (2–9 TA sites) rather than larger ORFs (TA sites >= 10) (*DeJesus et al., 2017*). There are 4136 (6.43% of total ORFs) nonpermissive TA sites in ATCC13950, which is less than in *M. tuberculosis* H37Rv (9% of total ORFs) *DeJesus et al., 2017* and in *M. abscessus* ATCC19977 (8.1% of total ORFs) (*Rifat et al., 2021*). As for small ORFs (2–9 TA sites), there are non-permissive TA sites in 41 genes (ORFs) of common essential or growth-defect-associated (1.35% of a total of 3021 larger ORFs in ATCC13950). With respect to the covering of non-permissive sites, our TnSeq data may need to be improved to classify quite accurately the gene essentiality, particularly on the nonstructural small regulatory genes, including small RNAs, although the study by DeJesus shows the comparable number of essential genes in large genes possessing more than 10 TA sites between 2 and 14 TnSeq datasets, most of which seem to be structural genes (*DeJesus et al., 2017*).

We have provided the data suggesting the preferential hypoxic adaptation in clinical strains compared to the ATCC type strain by the growth assay of individual strains. To strengthen our claim, several experiments are suggested, including mixed culture experiments of clinical and reference strains under hypoxia. However, co-culture conditions introduce additional variables, including inter-strain competition or synergy, which can obscure the specific contributions of hypoxic adaptation in each strain. Therefore, we took the current approach using monoculture growth curves under defined oxygen conditions, which offers a clearer interpretation of strain-specific hypoxic responses. Furthermore, one of the limitations of this study is the lack of validation of TnSeq results with individual gene knockouts. Contrary to the case of Mtb, the technique of constructing knockout mutants of slow-growing NTM, including *M. intracellulare* has not been established for a long time. We have just recently succeeded in constructing the vector plasmids for making knockout mutants of *M. intracellulare* (*Tateishi et al., 2024*). A growth assay of individual knockout strains of genes showing increased genetic requirements, such as *pckA*, *glpX*, *csd*, *eccC5*, and *mycP5* in the clinical strains is suggested to provide the direct involvement of these genes on the preferential hypoxic adaptation in clinical strains. We have a future plan to construct knockout mutants of these genes to confirm the involvement of these genes in preferential hypoxic adaptation.

In summary, we have identified 131 universal essential and growth-defect-associated genes that cover the genomic diversity of *M. intracellulare* strains responsible for MAC-PD, which narrowed down the drug targets against MAC-PD. Furthermore, we have demonstrated preferential adaptation to hypoxia in clinical NTM strains compared with the type strain in terms of the increased genetic

requirements for hypoxic growth, such as those encoding phosphoenolpyruvate carboxykinase, the ESX-5 type VII secretion system, and cysteine desulfurase. These genes were shown to have a significant impact on infection in vivo by increasing their genetic requirements. Our data highlight the importance of using clinical strains and hypoxic conditions in experimental research and provide fundamental resources for developing novel drugs directed towards nontuberculous mycobacterial strains that cause severe clinical diseases.

## Methods

### Bacterial strains and phages used to prepare Tn mutants

A total of eleven strains were used in this study, including nine clinical strains isolated from patients with MAC-PD in Japan (M.i.198, M.i.27, M018, M003, M019, M021, M001, M005, M016) (*Tateishi et al., 2023*), *M. intracellulare* genovar *paraintracellulare* type strain MOTT64 isolated from MAC-PD in South Korea (*Kim et al., 2012*) and the *M. intracellulare* type strain ATCC13950 (*Figure 1*, *Supplementary file 1*). All eleven clinical strains from MAC-PD patients in Japan were isolated from sputum (*Tateishi et al., 2021*; *Tateishi et al., 2023*). Sputum samples were treated by the standard method for clinical isolation of mycobacteria with 0.5% (w/v) N-acetyl-L-cysteine and 2% (w/v) sodium hydroxide and plated on 7H10/OADC agar. Single colonies were picked up for use in experiments as isolated strains. Of these strains, ATCC13950, M.i.198, M.i.27, M018, M005, and M016 belong to the typical *M. intracellulare* (TMI) genotype and M001, M003, M019, M021, and MOTT64 belong to the *M. paraintracellulare-M. indicus pranii* (MP-MIP) genotype (*Figure 1*, *Supplementary file 1*). The Tn mutant libraries were constructed using glucose as a carbon source under aerobic conditions according to the transduction method used for ATCC13950 (*Tateishi et al., 2020*), except that a higher concentration of kanamycin (100 µg/ml) was used to select the transposon mutants than was used for ATCC13950 (25 µg/ml). In brief, a hundred milliliter of the bacteria were cultured aerobically in Middlebrook 7H9 medium (Difco Laboratories, Detroit, MI, USA) containing 0.2% glycerol, 0.1% Tween 80 (MP Biochemicals, Illkirch, France), and 10% Middlebrook albumin–dextrose–catalase (7H9/ADC/Tween 80) until mid-log phase (optical density at a wavelength of 600 nm [$OD_{600}$] of approximately 0.4–0.6). After harvesting the bacteria, 10 mL of high-titer mycobacteriophage phAE180 (*Kriakov et al., 2003*) was transduced into the bacteria for 1 day at 37 °C for inserting Tn and plated on Middlebrook 7H10 solid medium supplemented with 10% oleic acid-albumin-dextrose-catalase (Becton Dickinson and Co., Sparks, MD) (7H10/OADC) agar plates containing kanamycin. After 2 or 3 weeks of cultivation at 37 °C aerobically, the colonies were harvested *en masse* and used as Tn mutant libraries. The kanamycin-resistant colonies (>1×10⁵) were harvested and evenly resuspended in 7H9/ADC medium containing 20% glycerol, aliquoted, and stored at −80 °C until further use.

### Next-generation sequencing and TnSeq analysis

DNA libraries were prepared as reported previously (*Minato et al., 2019*). The TnSeq libraries were sequenced with the NextSeq 500 System, 150 bp single-end run (Illumina, San Diego, CA, USA). The sequence reads were analyzed as previously described (*Tateishi et al., 2020*). Genetic requirements were calculated, and gene essentiality was predicted by resampling and HMM analyses using TRANSIT, respectively (*DeJesus et al., 2015*). First, an HMM analysis was conducted to calculate the fluctuation of the number of Tn insertion reads individually for each strain. The HMM method can categorize the gene essentiality throughout the genome, including 'Essential,' 'Growth-defect,' 'Non-essential,'and 'Growth-advantage.' 'Essential' genes are defined as no insertions in all or most of their TA sites. 'Non-essential' genes are defined as regions that have usual read counts. 'Growth-defect' genes are defined as regions that have unusually low read counts. 'Growth-advantage' genes are defined as regions that have unusually high low read counts. Universal essential and growth-defect-associated genes were identified as the genes classified as 'Essential' and 'Growth-defect' with an HMM analysis in all 9 subject strains (*Supplementary file 4*). Next, a resampling analysis was conducted to calculate the difference of the number of Tn insertion reads between each clinical strain and ATCC13950 (*Supplementary file 8*). By comparing the data obtained by HMM and resampling analyses, we summarized the difference in genetic requirements between the clinical MAC-PD strains and ATCC13950. To compare the genetic requirements of the clinical MAC-PD strains and ATCC13950, the annotation for the clinical MAC-PD strains was adapted from that of ATCC13950 by

adjusting the START and END coordinates of each open reading frame (ORF) in the clinical MAC-PD strains according to their alignment with the corresponding ORFs of ATCC13950. The number of Tn insertions in our datasets varied between 1.3–5.8 million among strains. To reduce the variation in the Tn insertion across strains, we adopt a non-linear normalization method, Beta-Geometric correction (BGC). BGC normalizes the datasets to fit an 'ideal' geometric distribution with a variable probability parameter $\rho$ (*DeJesus et al., 2015*), and BGC improves resampling by reducing the skew. On the TRANSIT software, we set the replicate option as Sum to combine read counts. And we normalized the datasets by BGC and performed resampling analysis by using the normalized datasets to compare the genetic requirements between strains. The function of TnSeq-hit genes was categorized according to the Kyoto Encyclopedia of Genes and Genomes (KEGG) database.

## Animals

Female C57BL/6JJcl mice were purchased from CLEA Japan (Tokyo, Japan). Six-week-old mice were used for the infection experiment (*Figure 6—figure supplement 1*). All the mice were kept under specific-pathogen-free conditions in the animal facility of Niigata University Graduate School of Medicine, according to the institutional guidelines for animal experiments. The mice were housed (four per cage) under a 12 hr/12 hr light/dark cycle. Food and water were available ad libitum. Kanamycin was added to the drinking water to a final concentration of 100 µg/ml to maintain the mice's resistance to the Tn mutant strains. Euthanasia was conducted with an intraperitoneal injection of a combined anesthetic (15 µg of medetomidine, 80 µg of midazolam, and 100 µg of butorphanol), followed by cervical dislocation.

## Mouse TnSeq experiment

The protocol was based on our previous mouse lung infection experiment (*Tateishi et al., 2023*). Twenty mice each were used for the Tn mutant strains M.i.198 and M.i.27. The mice were injected intratracheally with $3.25\times10^8$ and $5.93\times10^7$ CFUs of the Tn mutant strains, respectively. The infected lungs were expected to be harvested from five mice at day 1, week 4, week 8, and week 16 after infection. However, the lungs were harvested from only four mice infected with the M.i.27 Tn mutant strains in week 8 because we experienced difficulties with sampling, and the lungs were harvested from only three, one, and two mice infected with M.i.198 Tn mutant strains in week 8, week 12, and week 16, respectively, because morbidity was high in the mice infected with M.i.198. The harvested lungs were homogenized in 4.5 ml of distilled water with the gentleMACS Tissue Dissociator (Miltenyi Biotec, Bergisch Gladbach, Germany) and the homogenates were plated on 7H10/OADC agar plates and cultured for 3 weeks. The colonies were harvested *en masse* for TnSeq.

## Gene set enrichment analysis

To investigate the overlap of the gene sets required for growth in mouse lungs with those required for hypoxic pellicle formation observed in ATCC13950 and those showing increased genetic requirements observed in the clinical MAC-PD strains, a statistical enrichment of gene sets was performed by GSEA 4.3.2 (*Subramanian et al., 2005*). Parameters set for GSEA were as follows: permutations = 1000, permutation type: gene_set, enrichment statistic: weighted, metric for ranking genes: Signal-2Noise, max size: 500, min size: 15.

## Construction and growth assay of knockdown strains

Knockdown strains were constructed with the CRISPR-dCas9-mediated RNA interference system, as reported previously (*Singh et al., 2016*). Gene-specific 20-nt sequences were selected as protospacers for incorporation into the sgRNAs (*Supplementary file 20*). pRH2502 (a vector expressing an inactivated version of the *cas9* DNA sequence) and pRH2521 (an sgRNA-expressing vector whose expression is regulated by the TetR-regulated smyc promoter $P_{myc1}tetO$) were introduced into the bacteria to obtain colonies of the knockdown strains. For selection, 25 µg/ml kanamycin and 50 µg/ml hygromycin were used for ATCC13950, and 100 µg/ml kanamycin and 50 µg/ml hygromycin were used for the clinical strains. The growth assay of the knockdown strains was performed in 7H9/ADC broth containing the above-mentioned concentrations of kanamycin and hygromycin under aerobic conditions at 37 °C with humidification (APM-30D incubator, Astec, Tokyo, Japan). During the assay, the cultures were supplemented with anhydrotetracycline (aTc) to a final concentration of 50 ng/ml (as

a concentration that is not toxic to ATCC13950 and M.i.27) or 200 ng/ml (for the other clinical strains). aTc was added repeatedly every 48 hr to maintain the induction of dCas9 and sgRNAs in experiments that extended beyond 48 h (*Singh et al., 2016*). An aliquot (4 µl) of the culture and its diluents were streaked onto 7H10/OADC agar plates containing kanamycin and hygromycin, and the CFUs were counted. Growth inhibition was evaluated as the ratio of the CFUs of knockdown strains compared with those of the vector control strains lacking the guide RNA sequence. To confirm the downregulation of gene expression in the knockdown strains, quantitative reverse transcription PCR (qRT-PCR) was performed. Bacteria were cultured in 10 ml of 7H9/ADC/Tween 80 containing kanamycin and hygromycin until mid-log phase ($OD_{600}$ approximately 0.4–0.6). The cultures were supplemented with aTc every 48 hr, as in the growth assay for the knockdown strains. One day after the second addition of aTc, RNA was extracted by bead-beating six times at 5000 rpm for 30 s with cooling (Micro Smash MS 100 R Cell Disruptor; Tomy Seiko, Tokyo, Japan) and cDNA was synthesized with the Direct-zol RNA Miniprep (Zymo Research, Orange, CA, USA) and ReverTra Ace qPCR RT Master Mix with gDNA Remover (Toyobo, Osaka, Japan). qRT-PCR was performed with the CFX Connect Real-time PCR Detection System (Bio-Rad, Hercules, CA, USA), according to the manufacturer's instructions. The data were normalized to the expression of *sigA*.

## Comparison of growth curves under aerobic and hypoxic conditions

First, we precultured the bacteria in 7H9/ADC broth until the bacteria reached early log phase ($OD_{600}$ approximately 0.1–0.3). Next, the bacterial cultures were diluted with new 7H9/ADC broth (final $OD_{600}$ 0.003) for the aerobic and hypoxic growth assay. Bacterial cells were grown statically in 96-well plates containing 250 µl of 7H9/ADC broth in each well under aerobic conditions or 5% oxygen at 37 °C with humidification (APM-30D incubator, Astec, Tokyo, Japan). The culture was sampled every 1–3 days. At sampling, the culture was mixed by pipetting 50 times. An aliquot (4 µl) of the culture and its diluents were streaked onto 7H10/OADC agar and the CFUs counted. The growth curve was fitted to the logistic 4 P model with the JMP software (SAS Institute Inc, Cary, NC, USA) to calculate the inflection point (midpoint) and growth rate at midpoint in the log growth phase. Entry into the log phase was evaluated from the timing of the inflection point on the growth curve.

## Statistical analyses

The cut-off of fold change ($log_2FC$) to compare genetic requirements by TnSeq resampling analysis was set as adjusted *p*-values less than 0.05. Pearson's correlation test was performed between the fitness costs ($log_2FC$ calculated by the resampling assay), efficiency of knockdown (assessed by qRT-PCR) and growth suppression in the knockdown strains. Statistical analyses of the growth curves were performed using a 4-parameter logistic model (logistic 4 P model), Wilcoxon two-sample test, and a one-way analysis of variance (ANOVA) followed by Steel's multiple comparisons test using the JMP software program (SAS Institute Inc, Cary, NC, USA). Data are presented as the mean ± SD. Statistical significance was set as *p* values less than 0.05.

## Consent for publication

According to the protocols approved by the Institutional Review Board of Niigata University mentioned above, consent for publication was obtained from the patients enrolled in this study.

## Acknowledgements

The authors thank Dr. Torin Weisbrod and Dr. William R Jacobs, Jr. (Albert Einstein College of Medicine, Bronx, NY, USA) for kindly providing mycobacterial phage phAE180, and Dr. Norikazu Hara and Dr. Takeshi Ikeuchi (Department of Molecular Genetics, Bioresource Science, Brain Center for Bioresources, Brain Research Institute, Niigata University) for next-generation sequencing with the NextSeq 500 System. The authors thank Mr. Naomi Yoshihara and Ms. Chisa Kumagai for constructing Tn mutant library strains and hypoxic growth experiments, respectively. This work was supported by Grants-in-Aid for Scientific Research (grant numbers 20KK0216, 21K19637, and 22H03115 to Yoshitaka Tateishi) from the Ministry of Health, Labour, and Welfare, of Japan. This work was also supported by grants from the Japanese Ministry of Education, Culture, Sports, Science, and Technology, the Ministry of Health, and the Research Program on Emerging Infectious Disease from the Japan Agency

for Medical Research and Development (23fk0108673h0501, AMED-CREST 25gm1610009h0004, and JP223fa627005).

## Additional information

### Funding

| Funder | Grant reference number | Author |
|---|---|---|
| Ministry of Health Labour and Welfare | 20KK0216 | Yoshitaka Tateishi |
| Ministry of Health Labour and Welfare | 21K19637 | Yoshitaka Tateishi |
| Ministry of Health Labour and Welfare | 22H03115 | Yoshitaka Tateishi |
| Ministry of Education, Culture, Sports, Science and Technology | 23fk0108673h0501 | Sohkichi Matsumoto |
| Japan Agency for Medical Research and Development | 25gm1610009h0004 | Sohkichi Matsumoto |
| Japan Agency for Medical Research and Development | JP223fa627005 | Sohkichi Matsumoto |
| Japan Agency for Medical Research and Development | JP23gm1610013 | Yusuke Minato |
| Japan Society for the Promotion of Science | 23K27325 | Yusuke Minato |
| Japan Society for the Promotion of Science | 24K1163 | Yusuke Minato |

The funders had no role in study design, data collection and interpretation, or the decision to submit the work for publication.

### Author contributions

Yoshitaka Tateishi, Conceptualization, Resources, Data curation, Software, Formal analysis, Funding acquisition, Validation, Investigation, Visualization, Methodology, Writing – original draft, Project administration, Writing – review and editing; Yuriko Ozeki, Akihito Nishiyama, Resources; Yuta Morishige, Resources, Writing – review and editing; Yusuke Minato, Anthony David Baughn, Resources, Methodology, Writing – review and editing; Sohkichi Matsumoto, Resources, Supervision, Funding acquisition, Project administration, Writing – review and editing

### Author ORCIDs

Yoshitaka Tateishi ⓘ https://orcid.org/0000-0003-1200-0000
Yuriko Ozeki ⓘ http://orcid.org/0000-0003-0672-1039
Akihito Nishiyama ⓘ https://orcid.org/0000-0003-4416-7710
Yuta Morishige ⓘ https://orcid.org/0000-0002-3818-5288
Yusuke Minato ⓘ https://orcid.org/0000-0002-0888-8564
Anthony David Baughn ⓘ https://orcid.org/0000-0003-1188-4238
Sohkichi Matsumoto ⓘ http://orcid.org/0000-0002-6106-7538

### Ethics

Human subjects: All experimental protocols were approved by the Institutional Review Board of Niigata University (approval numbers: 2019-0020 for the use of human tissue samples). All experiments on humans and human samples were conducted in accordance with relevant guidelines and regulations. Informed consent was obtained from all patients.

All experimental protocols were approved by the Institutional Review Board of Niigata University (SA00506 for experiments on live vertebrates). All experiments on animals were conducted in accordance with the relevant guidelines and regulations, and the study complied with the ARRIVE guidelines.

Reviewer #1 (Public review): https://doi.org/10.7554/eLife.99426.5.sa1
Author response https://doi.org/10.7554/eLife.99426.5.sa2

## Additional files

### Supplementary files

Supplementary file 1. Genotypes of study strains and the strains used for each experiment.

Supplementary file 2. Number of reads obtained and TA coverage in each replicate of transposon (Tn) mutant library strains. Data for ATCC13950 are from our previous study (*Tateishi et al., 2020*).

Supplementary file 3. Number of essential, growth-defect-associated, nonessential, and growth-advantage-associated genes detected with an hidden Markov model (HMM) analysis in each *M. intracellulare* strain. Data for ATCC13950 are from our previous study (*Tateishi et al., 2020*).

Supplementary file 4. List of genes identified as universal essential or growth-defect-associated among the nine *M. intracellulare* strains analyzed in this study. ES: essential, GD: growth-defect-associated.

Supplementary file 5. List of genes identified as accessory essential or growth-defect-associated in the accessory genome and strain-dependent essential or growth-defect-associated in the core genes of *M. intracellulare*. 'Hit' represents the genes identified as essential or growth-defect-associated in each strain. The data of the pan-genome were referred to our previous study (*Tateishi et al., 2021*).

Supplementary file 6. Result of the gene set enrichment analysis (GSEA) in strain-dependent/accessory essential or growth-defect-associated genes. Genes required for hypoxic pellicle formation observed in ATCC13950 were ordered by their position in the ranked list of genes. Only 31 genes were ranked because the rest of the 144 genes were not present in the gene set of strain-dependent or accessory essential genes.

Supplementary file 7. Classification of the genes showing increased genetic requirements in the clinical *Mycobacterium avium–intracellulare* complex pulmonary disease (MAC-PD) strains compared to ATCC13950, and the genes of gluconeogenesis, fructose-1,6-bisphosphatase *glpX,* with respect to the core and accessory genomes.

Supplementary file 8. List of genes showing significantly increased or reduced transposon (Tn) insertion reads in the clinical *M. intracellulare* strains compared with ATCC13950 by a resampling analysis, with information of the genetic requirements detected by an hidden Markov model (HMM) analysis.

Supplementary file 9. Number of transposon (Tn) insertion reads in samples from infected mouse lungs. Individual mice are represented as A–E after the sampling time points.

Supplementary file 10. List of genes identified with a resampling analysis in mouse lungs infected with M.i.198 transposon (Tn) mutant library strains.

Supplementary file 11. List of genes identified with a resampling analysis in mouse lungs infected with M.i.27 transposon (Tn) mutant library strains.

Supplementary file 12. List of genes required for mouse lung infection that are also required for hypoxic pellicle formation by ATCC13950.

Supplementary file 13. List of genes required for mouse lung infection at all time points from Day 1 to Week 16, and from Week 4 to Week 16.

Supplementary file 14. List of genes required for mouse lung infection at all time points from Day 1 to Week 16, and from Week 4 to Week 16, as well as required for hypoxic pellicle formation in ATCC13950.

Supplementary file 15. Result of the gene set enrichment analysis (GSEA). Genes in the gene sets required for hypoxic pellicle formation observed in ATCC13950 and those showing increased genetic requirements observed in the clinical *Mycobacterium avium–intracellulare* complex pulmonary disease (MAC-PD) strains were ordered by their position in the ranked list of genes.

Supplementary file 16. Result of the gene set enrichment analysis (GSEA) in mouse Transposon sequencing (TnSeq) datasets. Genes required for hypoxic pellicle formation observed in ATCC13950 were ordered by their position in the ranked list of genes.

Supplementary file 17. Result of the gene set enrichment analysis (GSEA) in mouse Transposon sequencing (TnSeq) datasets. Genes showing increased genetic requirements observed in the clinical *Mycobacterium avium–intracellulare* complex pulmonary disease (MAC-PD) strains were ordered by their position in the ranked list of genes.

Supplementary file 18. Correlation analysis between the fitness costs (log$_2$FC), efficiency of knockdown (quantitative reverse transcription PCR, qRT-PCR) and growth suppression in the knockdown strains of strain-dependent/accessory essential or growth-defect-associated genes.

Supplementary file 19. The raw data of colony-forming units (CFUs) used for drawing growth curves in *Figure 8a*.

Supplementary file 20. Oligonucleotides and primers used in this study.

MDAR checklist

## Data availability

The TnSeq sequencing datasets are openly available in the DDBJ Sequence Read Archive database with the primary accession numbers DRA017301. The authors confirm that the data supporting the findings of this study other than the TnSeq sequencing datasets are available within the article.

The following dataset was generated:

| Author(s) | Year | Dataset title | Dataset URL | Database and Identifier |
|---|---|---|---|---|
| Tateishi Y | 2023 | Sequence read archives of M. intracellulare TnSeq in year 2023 | https://ddbj.nig.ac.jp/search/entry/sra-submission/DRA017301 | DDBJ Sequence Read Archive, DRA017301 |

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
