## [Editor Report · eLife Assessment]

This study makes a **valuable** contribution by elucidating the genetic determinants of growth and fitness across multiple clinical strains of Mycobacterium intracellulare, an understudied non-tuberculous mycobacterium. Using transposon sequencing (Tn-seq), the authors identify a core set of 131 genes essential for bacterial adaptation to hypoxia, providing a **convincing** foundation for anti-mycobacterial drug discovery.

---

## [Referee Report · Reviewer #1 (Public review)]

Summary:

In this descriptive study, Tateishi et al. report a Tn-seq based analysis of genetic requirements for growth and fitness in 8 clinical strains of Mycobacterium intracellulare (Mi), and compare the findings with a type strain ATCC13950. The study finds a core set of 131 genes that are essential in all nine strains, and therefore are reasonably argued as potential drug targets. Multiple other genes required for fitness in clinical isolates have been found to be important for hypoxic growth in the type strain.

Strengths:

The study has generated a large volume of Tn-seq datasets of multiple clinical strains of Mi from multiple growth conditions, including from mouse lungs. The dataset can serve as an important resource for future studies on Mi, which despite being clinically significant, remains a relatively understudied species of mycobacteria.

Weaknesses:

The primary claim of the study that the clinical strains are better adapted for hypoxic growth is yet to be comprehensively investigated. However, this reviewer thinks such an investigation would require a complex experimental design and perhaps form an independent study.

Comments on revisions:

The revised paper has satisfactorily addressed my previous concerns, and I have no further issues with this paper.

---

## [Author Response]

The following is the authors’ response to the previous reviews

**Reviewer #1 (Public review) :**
Comments on revisions:The revised manuscript has responded to the previous concerns of the reviewers, albeit modestly. The overemphasis on hypoxic adaptation of the clinical isolates persist as a key concern in the paper. The authors have compared the growth-curve of each of the clinical and ATCC strains under normal and hypoxic conditions (Fig. 8), but don't show how mutations in some of the genes identified in Tn-seq would impact the growth phenotype under hypoxia. They largely base their arguments on previously published results.As I mentioned previously, the paper will be better without over-interpreting the TnSeq data in the context of hypoxia.

Thank you for the comment on the issue of not determining the impact of individual gene mutations identified in TnSeq on the growth phenotypes under hypoxia.

We agree that the lack of validation of TnSeq results is a limitation of this study. Without evidence of growth pattern of each gene-deletion mutant under hypoxia there might be a risk of over-interpretating the data, even though the data are carefully interpreted based on previous reports. We consider that it is necessary to confirm the phenomenon by using knockout mutants.

We have just recently succeeded in constructing the vector plasmids for making knockout mutants of *M intracellulare* (Tateishi. Microbiol Immunol. 2024). We will proceed to the validation experiment of TnSeq-hit genes by constructing knockout mutants. We already mentioned this point as a limitation of this study in the Discussion (pages 35-36 lines 630-640 in the revised manuscript).

Reference.

Tateishi, Y., Nishiyama, A., Ozeki, Y. & Matsumoto, S. Construction of knockout mutants in *Mycobacterium intracellulare* ATCC13950 strain using a thermosensitive plasmid containing negative selection marker *rpsL+*. *Microbiol Immunol* 68, 339-347 (2024).

Other points:The y-axis legends of plots in Fig.8c are illegible.

Following the comment, we have corrected Figure 8c and checked the uploaded PDF

The statements in lines 376-389 are convoluted and need some explanation. If the clinical strains enter the log phase sooner than ATCC strain under hypoxia, then how come their growth rates (fig. 8c) are lower? Aren't they expected to grow faster?

Thank you for the comment on the interpretation of the difference in bacterial growth under hypoxia between MAC-PD strains and the ATCC type strain. The growth curve consists of the onset of logarithmic growth and its growth speed. In this study, we evaluated the former as timing of midpoint and the latter as growth rate at midpoint. Timing of midpoint and growth rate at midpoint are individual parameters. The early entry to log-phase does not mean the fast growth rate at midpoint.

Our results demonstrated that 5 (M.i.198, M.i.27, M003, M019 and M021) out of 8 clinical MAC-PD strains entered log-phase early and continued to grow logarithmically long time (slow growth). This data suggests the capacity for MAC-PD to continue replication long time under hypoxic conditions. By contrast, the ATCC type strain showed delayed onset of logarithmic growth caused by long-term lag phase. The duration of logarithmic growth was short even once after it started. The log phase soon transited to the stationary phase. This data suggests the lower capacity for the ATCC strain to continue replication under hypoxic conditions.

Following the comment, we have added the interpretation of the growth curve pattern as follows (page 22 lines 379-392 in the revised manuscript): “The growth rate at midpoint under hypoxic conditions was significantly lower in these 5 clinical MAC-PD strains than in ATCC13950. The early entry to log phase followed by long-term logarithmic growth (slow growth rate at midpoint) suggests the capacity for these 5 clinical MAC-PD strains to continue replication long time under hypoxic conditions. On the other hand, the rest 3 clinical MAC-PD strains (M018, M001 and MOTT64) did not show significant change in the growth rate between aerobic and hypoxic conditions, suggesting that there are different levels of capacity in maintaining long-term replication under hypoxia among clinical MAC-PD strains. In ATCC13950, the entry to log phase was significantly delayed under 5% oxygen compared to aerobic conditions, and the growth rate at midpoint was significantly increased under hypoxic conditions compared to aerobic conditions in ATCC13950. Such long-term lag phase followed by short-term log phase suggests lower capacity for ATCC13950 to continue replication under hypoxic conditions compared to clinical MAC-PD strains.”

**Reviewer #4 (Public review):**
Comments on revisions:The revised version has satisfactorily addressed my initial comments in the discussion section.

The authors thank the Reviewer for understanding our reply.

**Reviewer #5 (Public review):**
Comments on revisions:There is quite a lot of data and this could have been a really impactful study if the authors had channelized the Tn mutagenesis by focusing on one pathway or network. It looks scattered. However, from the previous version, the authors have made significant improvements to the manuscript and have provided comments that fairly address my questions.

The authors thank the Reviewer for understanding our reply. And the authors thank the Reviewer for the comments suggesting the future studies of TnSeq that focus on one pathway or network.